# Mechanisms of Cisplatin-Induced Acute Kidney Injury: Pathological Mechanisms, Pharmacological Interventions, and Genetic Mitigations

**DOI:** 10.3390/cancers13071572

**Published:** 2021-03-29

**Authors:** Kristen Renee McSweeney, Laura Kate Gadanec, Tawar Qaradakhi, Benazir Ashiana Ali, Anthony Zulli, Vasso Apostolopoulos

**Affiliations:** Institute for Health and Sport, Victoria University, Werribee, VIC 3030, Australia; kristen.mcsweeney@live.vu.edu.au (K.R.M.); laura.gadanec@live.vu.edu.au (L.K.G.); tawar.qaradakhi@live.vu.edu.au (T.Q.); benazir.ali@live.vu.edu.au (B.A.A.)

**Keywords:** cisplatin, acute kidney injury, AKI, cisplatin-induced acute kidney injury, nephrotoxicity

## Abstract

**Simple Summary:**

Nephrotoxicity is the dose-limiting factor of cisplatin treatment. Nephrotoxicity is characterized by reduced kidney function. Although an often-reversible condition, effects are notably seen years after treatment with cisplatin has ceased. It has an extensive pathophysiological map. The purpose of this article is to consolidate cisplatin-induced acute kidney injury literature and present it in one collective paper. It explores each individual mechanism linked to the disease, the pharmacological options that have been tested to target each of them, and the results obtained by each study. The paper also describes genetic modification studies and their effectiveness in preventing disease development.

**Abstract:**

Administration of the chemotherapeutic agent cisplatin leads to acute kidney injury (AKI). Cisplatin-induced AKI (CIAKI) has a complex pathophysiological map, which has been linked to cellular uptake and efflux, apoptosis, vascular injury, oxidative and endoplasmic reticulum stress, and inflammation. Despite research efforts, pharmaceutical interventions, and clinical trials spanning over several decades, a consistent and stable pharmacological treatment option to reduce AKI in patients receiving cisplatin remains unavailable. This has been predominately linked to the incomplete understanding of CIAKI pathophysiology and molecular mechanisms involved. Herein, we detail the extensively known pathophysiology of cisplatin-induced nephrotoxicity that manifests and the variety of pharmacological and genetic alteration studies that target them.

## 1. Introduction

### 1.1. Cisplatin

Cisplatin (cis-diamminedichloroplatinum II) is a platinum-containing antineoplastic drug first approved for clinical use in 1978 [1]. It is used extensively to treat a repertoire of malignancies per se or as a tailored combination in treatment [1]. Cisplatin is used to treat breast [2], cervical [2], oesophageal [3], bladder [4], small cell lung [5], and testicular cancers [6]. Cisplatin is also used as a combination therapy to treat high grade cancers such as osteosarcoma [7] and soft-tissue cancers including squamous cell carcinoma [8]. Cisplatin is one of the most potent and effective chemotherapies used to date [9], and its antitumor effects are well established [2,9,10]. However, the exact mechanism of cisplatin-induced cell death remains largely unknown. It is widely accepted that cisplatin causes 1–2 intrastrand or 1–3 interstrand crosslinks with purine bases on the deoxyribonucleic acid (DNA) strand [9,11]. This crosslinking impairs DNA repair mechanisms, inhibiting the production of a viable DNA replication template, stimulating cell-cycle arrest leading to cell death [12]. Irrespective of its potent anticancer properties and efficacy, the clinical usage of cisplatin is limited due to the severity of adverse side effects including ototoxicity and neurotoxicity [13,14] and its dose-limiting factor nephrotoxicity [12,15,16,17,18,19,20,21]. 

### 1.2. Nephrotoxicity 

Nephrotoxicity results from a rapid decline of excretory mechanisms within the kidney [22], enhancing the accretion of waste products produced by protein metabolism (including urea, nitrogen, and creatinine) [22,23,24]. Acute kidney injury (AKI) is commonly caused by nephrotoxic injury to kidney tissue, resulting in acute tubular necrosis [25]. It can also result from inadequate urinal drainage [26]. Decreased drainage causes an increase in intratubular pressure and decreases glomerular filtration rate (GFR). Decreased GFR can additionally be stimulated by afferent arteriole vasoconstriction [27]. Despite improved prognosis following the removal of diuretics to promote volume expansion and hydration, the prevalence of cisplatin-induced AKI (CIAKI) remains high [28]. Although cisplatin-induced nephrotoxicity can manifest in a variety of ways, acute tubular necrosis (ATN) is the most prevalent [29]. In the clinical setting, AKI frequently occurs despite low-dose cisplatin administration [30]. The uptake of cisplatin into proximal tubular epithelial cells (PTEC) is the initiator of the toxic effects of cisplatin [31]. To date, despite burgeoned research, there is no intervention that adequately treats or prevents CIAKI in cancer patients [32]. Therefore, further understanding the molecular pathways and their interactions is essential in finding or developing a suitable pharmacological treatment to be used in conjunction with cisplatin.

### 1.3. Pathophysiology of Cisplatin-Induced AKI 

A variety of molecular pathways and mechanisms have been investigated to determine the unknown pathological events caused by CIAKI. The key molecular mechanisms involved in cisplatin-induced nephrotoxic adverse effects include cellular uptake and accumulation, inflammation, oxidative stress, vascular injury, endoplasmic reticulum (ER) stress, and necrosis and apoptosis (Figure 1). A plethora of pharmacological agents (Table 1) and genetic alterations (Table 2) have been investigated in experimental preclinical studies of CIAKI. Despite the prevalence of nephrotoxicity in cisplatin-treated patients, its clinical application must be accompanied by other treatments to counteract its harmful effects while allowing it to exert its potent anticancer properties. Cisplatin cellular uptake is the initiator of the nephrotoxic effects, with several studies investigating the various therapeutic options that promote renoprotection (Figure 2). The purpose of this review is to collectively present the magnitude of preclinical studies in addition to presenting the clinical studies recently completed and currently being conducted for the treatment of CIAKI. The data from these studies illustrates the broad pathophysiological mechanisms involved and the potential for their inter-relationships. This review sheds light on the current failure in preclinic to clinic translatability given the lack of studies currently moving from animal models to human clinical trials. Despite the frequent protective therapies evaluated in models of CIAKI, there is no evidence of treatment progression with almost all therapies evaluated, posing a highly concerning issue for cisplatin patients. 

## 2. Pharmacological Approaches Targeting Cisplatin Cellular Uptake

### 2.1. Cellular Uptake Transporters of Cisplatin 

The cellular uptake of cisplatin has been implicated in the pathogenesis of CIAKI. Organic cation transporter 2 (OCT2), copper transporter 1 (CTR1), and the less explored volume-regulated anion channels (VRAC) are involved in cisplatin transportation into kidney cells [57] by enabling platinum accumulation, which has been linked to kidney dysfunction [28] (Figure 2). Kidney tissue following cisplatin treatment showed a five-fold increase in cisplatin concentration compared to serum, indicative of PTEC accumulation [9]. Organic cation transporter 2 is one of the transporters affiliated with cisplatin cellular uptake.

### 2.2. Organic Cation Transporter 2 (OCT2) 

OCT2 is expressed on the basolateral membrane of PTEC [57,59,60] and plays a central role in cisplatin uptake into tubular cells [57,59,60]. Amongst the transporters responsible for CIAKI, it has been shown that 30% of nephrotoxic effects caused by cisplatin is directly mediated by OCT2 uptake [68]. In in vitro studies investigating OCT2-mediated cisplatin cellular uptake, pharmacological inhibition was noted of OCT2 by cimetidine-inhibited cisplatin-induced apoptosis. In addition, nephrotoxicity stimulated by cisplatin transportation into renal tubular cells and subsequent platinum accumulation could be decreased with orally administered imatinib (a tyrosine kinase inhibitor) in rats. Histological investigations of kidney tissue confirmed that there was no evidence of severe renal damage in mice co-treated with cisplatin and imatinib. However, tubular degeneration was observed in cisplatin-treated groups [61]. The results of blood analysis (plasma urea, nitrogen, creatinine, and creatinine clearance) were indicative of improved kidney function and platinum accumulation following imatinib adjunct therapy. In OCT2-expressed HEK293 cell studies, adjunct administration of cisplatin and imatinib showed decreased accumulation of platinum in PTECs and decreased cisplatin-induced cytotoxicity [61]. However, despite the renoprotective effects observed in preclinical animal models, imatinib has not provided positive toxicology results. According to the US Food and Drugs Administration adverse reporting system, 44 imatinib-treated cases cited renal-related toxicity. Of these 44 cases, 25 manifested as AKI [69]. As such, this may not be an adequate clinical treatment. Potentially irreversible acute kidney injury was also observed in a nonclinical trial in imatinib-treated chronic myeloid leukemia patients [70].

An experimental study using OCT2-deficient mice showed impairment of cisplatin uptake in renal cells, evident by reduced platinum accumulation [60]. Cairimboli et al. confirmed the importance of OCT2 in cisplatin uptake [58]. The authors associated the overexpression in HEK293 cells with increased cisplatin uptake causing cisplatin toxicity, because of increased cellular sensitivity [58]. To date, many pharmacological approaches targeting molecules responsible for cisplatin uptake or transportation into PTECs have been explored [57,61,71,72,73,74]. A murine model of CIAKI demonstrated downregulation of OCT2 expression by formononetin inhibited the development of AKI associated with cisplatin treatment through stimulation of renal tubular cell proliferation, survival, and apoptosis inhibition [71]. Despite the ameliorating effects of in vivo OCT2 inhibition in murine models of CIAKI, human studies failed to display the same renoprotective effects. Fox and colleagues used a randomized crossover experimental design to assess the prevention of cisplatin-induced nephrotoxicity using the OCT2 inhibitor pantoprazole, in young patients with osteosarcoma. To assess the effects, novel biomarkers were tested to investigate glomerular and tubular function. Measurement of serum cystatin c was used as an indirect indicator of GFR, and urinary biomarkers N-acetyl-β-glucosaminidase (NAG), kidney injury molecule-1 (KIM-1), and neutrophil gelatinase-associated lipocalin (NGAL) were used to quantify the degree of renal injury caused by cisplatin. The results of this study showed that concurrent administration of cisplatin with pantoprazole provided no protection against renal injury or function in young cancer patients [75]. Interestingly, a more recent study showed that pantoprazole can ameliorate CIAKI in mice [62]. Given the contradictory results of OCT2 inhibition on CIAKI in animal-versus-human studies further research needs to be conducted. It is important to note that OCT2 murine models were nontumor bearing, whilst the human studies were conducted in cancer patients, which could be a contributing factor to the failed clinical study. 

### 2.3. Copper Transporter 1

CTR1 is located on the basolateral membrane of proximal tubules and is highly expressed in human kidneys [57,59]. The exact role of CTR1 in cellular uptake of cisplatin into renal proximal tubules resulting in nephrotoxicity is incompletely understood. However, studies have shown that CTR1 downregulation is protective against platinum accumulation [57]. The knockdown of CTR1 reduces cisplatin nephrotoxicity by up to 80% in both mouse embryonic fibroblasts and yeast [72,74]. In vivo studies indicated elevated levels of CTR1 expression was associated with increased cisplatin accumulation in tumors, a process also observed in PTEC [57,76]. Pabla and colleagues investigated the relationship between cisplatin and CTR1 expression to further define the role that CTR1 plays in nephrotoxicity. Interestingly, in mice, there were no significant differences in CTR1 expression 1–3 days following cisplatin treatment. They also demonstrated that incubation of HEK293 cells with copper generated both monomeric and trimeric CTR1 knockdown, resulting in approximately 50% diminution in cisplatin accumulation and a 30% reduction in apoptosis. Furthermore, CTR1 knockdown cells incubated with the OCT/MATE inhibitor cimetidine further inhibited both cellular uptake and apoptosis following treatment with cisplatin [57,58]. The results of this study have shown that although both CTR1 inhibition and OCT2 inhibition alone are options to prevent nephrotoxicity, the combination of CTR1 and OCT2 inhibition together has better therapeutic potential. Interestingly, the majority of cellular uptake research regarding cisplatin into renal cells has focused on the two major cisplatin transporters OCT2 and CTR1. However, there have also been suggestions that there are other entry points for cisplatin into renal cells that are yet to be explored in models of CIAKI such as VRAC channels. Reduced VRAC channel activity is associated with cisplatin resistance [77], and the presence of VRAC channels in kidney cells [78] highlights a potential avenue for CIAKI research. Additionally, impaired cisplatin efflux has been shown to contribute to cisplatin accumulation and the nephrotoxic effects that follow [65,79]. 

### 2.4. OAT1/OAT3

In addition to OCT2 and CTR1, the organic anion transporter (OAT) family(OAT1 and OAT3 have also shown to transport cisplatin and potentially a nephrotoxic metabolite into PTEC resulting in nephrotoxic injury to renal cells [63]. OAT transporters are largely concentrated in the basolateral membrane of PTEC and facilitate transportation of hydrophilic anions into cells. This intake is via secondary/active transportation responsible for regulating anion balance in the body [80]. To investigate the influence of OAT transporters on cisplatin-induced nephrotoxicity, C57BL/6J mice with genetic deletion of OAT1 and OAT3 were injected with 30 mg/kg cisplatin. In cisplatin-treated wildtype mice, there were increases in biomarkers of CIAKI, in addition to histological indication of kidney damage such as tubule dilation and necrosis. There was no evidence of kidney dysfunction in OAT1- and OAT3-deficient mice treated with cisplatin. Further studies are needed to further understand the role of each individually and the interaction they have together on CIAKI nephrotoxicity. A different model was used to investigate OAT-stimulated CIAKI using nilotinib. Nilotinib is a tyrosine kinase inhibitor shown to noncompetitively inhibit OCT2 and both OAT1 and OAT3 [63]. Nilotinib was given to OCT1/2 ^−/−^ mice simultaneously with cisplatin, with results indicating no loss of kidney function in the adjuvant cisplatin- and nilotinib-treated group as confirmed by reduced BUN levels. This indicates that OAT1/inhibition by nilotinib provides some evidence of amelioration of CIAKI; however, as the paper elucidates, further investigations in the mechanisms of mitigation are required [63]. A separate study investigated the effects of nilotinib in a rodent model of CIAKI. Male Wister albino rats were treated with 25 mg/kg Nilotinib 4 days prior to a single intraperitoneal injection of 6 mg/kg cisplatin and 6 days following the cisplatin injection. Results of their study showed that nilotinib improved creatinine clearance compared to cisplatin-treated rats; however, it had no influence on increased BUN [64]. It was also observed that nilotinib attenuated cisplatin increase in MDA, a biomarker of oxidative stress [81]. Morphological changes showed amelioration of CIAKI by nilotinib. Although this study did not look specifically at nilotinib influence on OAT1 and OAT3, it does confirm its ability to prevent CIAKI. Given this information, there is a clear correlation and link between the transporters, and therefore, further investigations need to be undertaken to understand their interactions and the influence that has on mediating cisplatin uptake. Given cisplatin uptake is the initial step mediating its nephrotoxic effects, potentially inhibiting all three synergistically may be an ideal strategy for CIAKI prevention. 

## 3. Pharmacological Approaches Targeting Cisplatin Cellular Efflux

Apically localized efflux transporters P-type copper transporting ATPases (ATP7A and ATP7B), multi-antimicrobial extrusion protein transporter-1 (MATE 1), and multidrug-resistance-associated protein (MRPs) mediate excretion of cisplatin into the urine [57,59,65,82]. These transporters are highly expressed in the proximal and distal tubules [57]. Tubular injury is a key pathology associated with the nephrotoxic effects of cisplatin. Tubular injury promotes reduced GFR and therefore delayed urinary excretion of cisplatin, leading to platinum accumulation within the tubules [83]. Given the pathogenesis linked to platinum accumulation in PTEC, increasing the expression of cisplatin efflux transporters has been a molecular target against CIAKI.

### 3.1. Apically Localized Efflux Transporters P-Type Copper Transporting ATPases7A/B 

Although there is little research available determining the effects of ATP7A and ATP7B on cisplatin nephrotoxicity, they have both been extensively investigated in the setting of cisplatin drug resistance. Studies investigating overexpression of both ATP7A and ATP7B in cancer models have been shown to be independently linked to poor survival in ovarian cancer patients and cisplatin resistance in prostate carcinoma cells, respectively [84,85,86]. Therefore, overexpression of ATPases could prevent CIAKI; however, its therapeutic potential may not exceed the possible negative outcomes to cancer cells.

### 3.2. Multidrug-Resistance-Associated Protein 2

MRPs are associated with mediating the efflux of cisplatin and its nephrotoxic conjugates from kidney cells [87]. Previously, research investigating the role of MRP expression in cisplatin accumulation demonstrated that increased MRP expression resulted in reduced cisplatin accumulation and therefore has been suggested to play a critical role in cisplatin-induced nephrotoxicity [87]. Given its role in nephroprotection, you would expect its expression to be downregulated in response to cisplatin treatment, given that platinum accumulation is a well-established complication. However, acute renal failure induced in rats showed a significant upregulation of MRP2 72 h after cisplatin treatment; however, there were only minor increases in MRP4 expression compared to controls [88]. 

The glutathione-s-platinum conjugate, whose metabolism is responsible for the production of the reactive thiol nephrotoxin, is suggested to be eliminated by MRPs [89]. To determine the role of MRP2 on cisplatin efflux, MRP2-deficient mice were treated with 20 mg/kg of cisplatin, resulting in enhanced platinum accumulation and proximal tubular injury. MRP2-deficient mice showed increased mRNA expression of GST, the enzyme which catalyzes the formation of the cisplatin–glutathione conjugate. Platinum accumulation was reduced in transgenic knock-in Mrp2-knockout mice, indicating that MRP2 plays a role in the accumulation of platinum [66]. However, the mechanisms involved in this accumulation and the effect they have on the production of the reactive thiol remain unclear. It is possible that an increase in MRP expression might be seen in models of nephrotoxicity; however, no studies are yet to present data investigating this. In addition to MRPs, multi-antimicrobial extrusion protein 1 (MATE1/SLC47A1) is suggested to be involved in platinum accumulation associated with cisplatin treatment [83].

### 3.3. Multi-Antimicrobial Extrusion Protein 1

MATE1 expression is largely concentrated in the brush-border membrane of PTECs and assists epithelial cell elimination of cationic molecules into urine [83]. Cisplatin has been identified as a substrate for MATE1 [90]. MATE1 has shown to mediate cisplatin efflux and prevent cisplatin accumulation in tubular cells preventing cisplatin nephrotoxic effects [65]. However, following cisplatin treatment, downregulation of MATE1 expression in human tubular epithelial cells is observed [83]. Given the ability for MATE1 to facilitate cisplatin urinary excretion, further investigations have been undertaken to isolate its role in cisplatin-induced AKI. A model of cisplatin-induced nephrotoxicity was undertaken in MATE1 ^−/−^ mice [65]. Blood urea nitrogen (BUN) and plasma creatinine concentration were significantly elevated in cisplatin-treated MATE1 ^−/−^ mice, compared with cisplatin-treated wildtype mice. It was observed that renal concentrations of platinum were at a 20-fold increase compared to plasma concentration in MATE1-deficient mice [65]. Increased serum creatinine (sCr) and BUN were also observed in MATE1 pharmacological inhibition studies [65,91]. Therefore, it has been concluded that MATE1 plays a key role in cisplatin accumulation and is thus a contributor to CIAKI [65]. A study investigating MATE1 upregulation in models of CIAKI may be a good therapeutic avenue in the prevention of nephrotoxicity. 

## 4. Interventions Targeting Molecular Mechanisms CIAKI

### 4.1. Oxidative Stress 

Despite the collection of research that has focused specifically on cisplatin transportation and accumulation, other models of CIAKI have targeted key molecules involved in ROS formation [92]. A balance occurs between ROS production and the antioxidant defense system to maintain homeostasis [2,3]. Cisplatin disrupts this equilibrium through overproduction of ROS and impaired antioxidant defense systems. This triggering reduced production of key antioxidants, including superoxide dismutase (SOD), glutathione (GSH) [2], and catalase (CAT) [93]. The reduction in the functionality of the antioxidant defense system leads to overexpression of key markers of oxidative stress following cisplatin treatment [36]. Elevated levels of cisplatin in PTEC also increase cisplatin accumulation in mitochondria, stimulating mitochondrial dysfunction, mitochondrial damage, and ROS production [94,95]. Following cisplatin infiltration into renal epithelial cells, it becomes a potent nephrotoxin via gamma-glutamyl transpeptidase (GGT)-dependent metabolic activation (Figure 2) [96]. Glutathione-S-transferase (GST) mediates the formation of glutathione-S-platinum conjugates, which passes through the kidneys [34]. It is cleaved into cysteine–glycine conjugate by Gama-glutamyl-transpeptidase [67]. Amino-dipeptidases further metabolizes this into cysteine conjugates, which are then transported into proximal tubules. Cysteine-S-conjugate β-lyase (CSCβL) is metabolized to form the cysteine conjugate into a highly reactive thiol, which is the initiator of cisplatin’s cell toxicity [67,96]. Elevated levels of the highly reactive thiol molecule produced after cisplatin uptake is metabolized by CSCβL catalyze the enzymatic activation of glucose-6-phosphate dehydrogenase and hexokinase, increasing ROS [1]. 

Studies have determined the use of natural antioxidants including, vitamin C [97], vitamin E [98], and activation of the vitamin D receptor [43] to target ROS formation, which have all shown nephroprotective properties against renal toxicity [15]. The beneficial co-therapy of cisplatin and vitamin C in C57BL/6 mice has been previously demonstrated. Mice were inoculated with Lewis lung carcinoma followed by treatment with cisplatin. Levels of sBUN and sCr presented cisplatin-treated mice demonstrated higher levels of oxidative damage. Decreased levels of kidney dysfunction were also observed in adjunct vitamin C and cisplatin-treated mice, without compromising cisplatin cytotoxicity [97]. Furthermore, there have been extensive pharmaceutical interventions assessed for their antioxidant effects on CIAKI [94]. Oxidative stress, evident by reduced MDA/MPO expression, was observed in necrostatin-1- and cisplatin-treated mice [99]. This was also observed in hesperetin- and cisplatin-treated HK-2 cells. Both drugs, showed reduced levels of apoptosis [100], which could interfere with the cytotoxicity of cisplatin. Experimental and clinical studies have focused on cisplatin nephrotoxicity, specifically through targeting mechanisms and molecular pathways associated with pharmacological inhibition ROS production or stimulation of antioxidant pathways [36,44,49,92,93,97,101]. A pathway of interest recently targeted in CIAKI research is the nuclear factor erythroid 2-related factor 2/heme oxygenase-1 (Nrf2/HO-1) signaling pathway [100].

#### Monotropein (Nrf2/HO-1 Antioxidant Pathway)

Activation of Nrf2 has shown promise in multiple experimental models as a key modulator in the suppression of oxidative stress and inflammation to preserve kidney function [102,103]. Nrf2 is responsible for maintenance of the cellular redox balance, antioxidant response, and phase II detoxification process [104]. Nrf2 expression has shown to be downregulated in rats following cisplatin treatment [105]. Renal expression of Nrf2 and HO-1 was downregulated compared to control rats, however rats pretreated with Sinapic acid (SA) followed by cisplatin resulted in marked increase in Nrf2 and Ho-1 expression. Cisplatin treatment resulted in a significant downregulation of key antioxidant enzymes SOD, CAT, and GSH. SA and cisplatin-treated rats showed elevation in these key antioxidants indicating an enhanced antioxidant defense system following SA treatment [103]. Pharmacologically, it has been demonstrated that activation of the Nrf2 signaling pathway by N,N-dimethylformamide (DMF) following cisplatin treatment attenuated AKI [102], as well as tubulointerstitial lesions [106]. Both ameliorating effects are associated with stimulation of antioxidants such as HO-1 and NAD(P)H quinone oxidoreductase 1 (NQO1) [102,106]. Stimulation of these two antioxidants was observed in mice treated with Isoorientin a flavone, suggested to activate the Nrf2 signaling cascade [44]. This was confirmed in Isoorientin-treated Nrf2-deficient mice, where renoprotection was abolished [44]. Taken together, these studies provide an insight to the promising effects of the Nrf2 pathway as a target for CIAKI. However, studies into the interaction between Nrf2 signaling activators and the apoptotic pathways that mediate cisplatin’s cytotoxicity have yet to be determined.

### 4.2. Vascular Injury 

Interestingly, a focus on anti-inflammatory and antioxidant pathways has been highlighted in most pharmaceutical and genetic modification studies published recently. Reduced GFR caused by reduced renal blood flow is a key pathology of CIAKI [107]. Little research specifically targeting the vasoconstriction properties of cisplatin to promote renal perfusion has been undertaken. Vasoconstriction stimulated through activation of adenosine A_1_ receptors (AT_1_s) by cisplatin is a suggested mechanism contributing to CIAKI. Additionally, CIAKI has been linked to vascular injury via endothelial dysfunction [31,107]. Reduced renal blood flow to kidney tissue through elevated vasoconstriction and impaired vascular autoregulation stimulated by damage to the endothelium is implicated in the pathogenesis of CIAKI [12]. Cisplatin has been suggested to cause damage to the vasculature within renal tubules [12]. It results in vascular resistance and constriction of vascular smooth muscle cells (VSMC) leading to reduced renal blood flow, decreased GFR, and hypoxia of renal tubular cells, leading to kidney damage [31,108]. Cisplatin has shown to alter the response of renal vascular endothelium to vasoactive substances [15]. Kidney vasculature and tubules are known to have extensive sympathetic nerve innervation, releasing catecholamines from their terminals, and triggering G-coupled adrenoceptors on the cell surface [109]. Adrenoceptors increase calcium and trigger contractions of vasculature muscle resulting in vasoconstriction of smooth muscle cells [110,111]. Vascular injury is linked to elevations in oxidative stress, resulting in a sequence of metabolic disturbances. Cisplatin treatment leads to oxidative stress induced by ROS and impaired function of the antioxidant defense system [79]. Excessively produced ROS include superoxide (O_2_^−^), hydroxyl radical (HO•), hydrogen peroxide (H_2_O_2_), peroxynitrite (ONOO^−^), nitrogen oxide (NO•), and hypochlorous acid (HOCl). Increased oxidative stress leads to endothelial damage, resulting in impaired endothelial dependent vascular relaxation. Endothelial dependent VSMC relaxation is NO dependent and alterations in its production or bioavailability disrupt relaxation. Additionally, eNOS function is impaired when ROS production or function is not repressed [112]. Amino acids such as L-arginine synthesize the production of eNOS to produce NO [9]. Excess production of NO by inducible NO synthase is increased following cisplatin treatment [64]. NO reacts with the NO scavenger O_2_^−^ to produce peroxynitrite (ONOO^−^) [4,10,11,12], resulting in reduced NO bioavailability and endothelial dysfunction [113].

Recently, levosimendan, a calcium sensitizing vasodilator, has been used in a model of CIAKI [114]. It attenuated renal damage, improved renal blood flow, and enhanced kidney morphology. Despite all those factors, there was only slight alleviation in biomarkers associated with kidney dysfunction (sCr and BUN), indicating only partial prevention of CIAKI. Levosimendan significantly reduced TNF-α expression in cisplatin-treated rats, indicating levosimendan also displays anti-inflammatory properties [114]. Interestingly, despite levosimendan renoprotective effects, earlier studies demonstrated it also plays a role in the prevention of H_2_O_2−_induced apoptosis of cardiomyocytes [115], indicating antiapoptotic properties taken together with levosimendan may potentially interfere with the cytotoxic properties of cisplatin. The vasculature plays a role in the pathophysiology of CIAKI; to further elucidate these effects, research into the renin angiotensin system (RAS) has been studied. 

#### The Renin Angiotensin System in Cisplatin-Induced Acute Kidney Injury

The RAS has been investigated in models of CIAKI for at least two decades to further identify the direct effects of cisplatin on the vasculature and renal hemodynamics [116]. The RAS is responsible for the homeostatic balance of blood pressure. Angiotensin I (Ang I) is converted to angiotensin II, which stimulates angiotensin II type I receptor (AT_1_) and angiotensin II type 2 receptor (AT_2_), stimulating vasoconstriction and vasodilation, respectively [117,118]. Stimulation of AT_1_ has been associated with renal injury [119], whilst AT_2_ activation has been correlated with renoprotection, through IL-10 stimulation and IL-6 downregulation [118,120,121,122]. Interestingly both receptors have been investigated in a CIAKI model to assess the role of both angiotensin II receptors in nephrotoxicity (Figure 3).

The role of the AT_1_ receptor in response to nephrotoxicity is conflicting amongst the literature, with evidence linking it to renoprotection in AT_1_ lymphocyte knockout models, whilst renal epithelial AT_1_ knockout worsened AKI pathogenesis [126]. Interestingly, it has been reported that AT_1_ stimulation worsened CIAKI through increased TNF-alpha [126]. Renal dysfunction and TNF-alpha expression was reduced in mice deficient in PTEC AT_1_ receptor expression compared to WT, highlighting the protective effects of the AT_1_ receptor in CIAKI [126]. Confirming these results, treatment of rats with the selective AT_1_ antagonist telmisartan almost restored BUN and sCr back to control levels indicative of restored renal function [124]. Interestingly telmisartan has shown in a mouse model of CIAKI to exacerbates cisplatin-induced nephrotoxicity [123]. 

These results show that there is both ameliorating effects between AT_1_ deficiency and stimulation in CIAKI. Additionally, this was supported by treatment with another AT1 receptor antagonist candesartan [127], with one study reporting no impact on reduction in BUN and creatinine following cisplatin treatment [125], whilst another showed protection. Given this information, further understanding of the role of the AT1 receptor in the pathogenesis of cisplatin-induced AKI is a critical component of understanding the pathophysiological map of the disease. 

Administration of telmisartan could be causing off-target effects given the renoprotective effects were attributed to antioxidant and anti-inflammatory properties rather than its involvement in the RAS [124]. The protective effects of AT_1_ inhibition in cisplatin-treated mice may be through increased activation of AT_2_ or the mitochondrial assembly receptor (MAS) receptor, resulting in enhanced vascular relaxation and promoting medullary blood flow. However, antagonism of the AT_2_ receptor by PD123319 improved renal function, indicated by reduced BUN and sCr when concurrently treated with cisplatin [125], suggesting that although stimulation of AT_2_ is vasoprotective, it has exhibited both renoprotective and renotoxic properties [125]. Given this information, a cisplatin model using a potent and highly selective AT_2_ receptor agonist may elucidate further as to its role in nephroprotection, specifically in CIAKI. 

Vascular dysfunction is worsened by the overactivation of AngII production and the depletion of Angiotensin converting enzyme 2 (ACE2) [128,129,130,131]. Angiotensin converting enzyme 2 (ACE2) activation stimulates production of angiotensin (1-7) from AngII metabolism [132], increasing vascular relaxation through activation of the MAS receptor [133]. Morsi et al. (2015) confirmed AngII plays a role in cisplatin-induced nephrotoxicity; however, the direct effects it has on vascular relaxation remains unclear. In cisplatin-treated rats, there was increased protein expression of AngII, iNOS, TNF-α, and caspase-3 and decreased expression of eNOS. Nebivolol itself had no impact on protein expression; however, adjunct treatment of nebivolol with cisplatin improved eNOS expression and reduced expression of AngII, iNOS, TNF-α, and caspase-3 compared to cisplatin. Nebivolol is a selective β1-adrenoreceptor antagonist, shown to have microvasculature vasodilatory [134], antioxidant [112,135], anti-inflammatory [136], and antiapoptotic properties. To determine the exact effects of nebivolol has on vascular function, further examinations into AT_1_ and AT_2_ with its use should be conducted. It is also unclear as to the exact mechanisms mediating the renoprotective effects of nebivolol, given that it may be having effects on the vasculature, inflammatory responses, and apoptotic pathways [64]. Further research specifically investigating the direct effects of vasodilation on cisplatin accumulation and resulting nephrotoxicity needs to be conducted. Cisplatin reduces expression of key molecules responsible for cisplatin efflux whilst increasing cellular uptake transporters. As such, specifically increasing blood flow without attenuating other pathologies such as inflammation and ROS production may in fact worsen CIAKI through increased platinum accumulation, potentiating worsened PTEC death. 

### 4.3. Cell Death

There are two mechanisms whereby cisplatin induces cell death, necrosis, and apoptosis. Initial research showed necrosis to be the only mechanism responsible for renal damage caused by cisplatin. However, Lieberthalet et al. (1996) were some of the first to show that apoptosis also played a part in cisplatin-induced cell death. The study showed that high-dose cisplatin treatment induced cell death via necrosis, whilst a low dose stimulated apoptotic cell death. Clinically cisplatin is administered through frequent low-dose infusion in attempts to prevent nephrotoxic effects, which differs from the previous high-dose method [137]. However, studies investigating the relationship between necrosis and apoptosis continues to be explored as death mechanisms caused by cisplatin-induced nephrotoxicity remain elusive [15].

#### 4.3.1. Necrosis 

Cisplatin induces cell death phenotypes in a concentration dependent manner. High concentrations of cisplatin cause a type of death independent of the classical features of apoptosis which resembles necrosis [138,139,140]. Necrotic damage is localized to the PTEC rather than the distal tubule as it reabsorbs filtered molecules including glucose, proteins, electrolytes, and drugs [141]. Sancho-Martinez et al. (2011) conducted a study in vitro using HK2 and Jurkat T cells, which showed activation of the apoptotic program with high necrotic concentrations of cisplatin [141]. The apoptotic program is further aborted at the level of effector caspases which emanates into necrotic-like death phenotype [141]. Necrosis is a passive mode of cell death which activates an inflammatory and immune response [141,142]. It is characterized by cell swelling, plasma membrane rupture, and loss of organelle structure [143]. Necrosis was considered as accidental cell death, which is unregulated [144] until genetic programmed necrosis was discovered in vivo [145,146]. A type of receptor-interacting protein kinase-based cell death that has similar signaling pathways with apoptosis and plays a major role in CI-AKI is namely necroptosis [142,147]. Xu et al. (2015) used RIP3-KO and mixed-linage kinase domain-like protein (MLKL) knockout mice to investigate the role of necroptosis in CIAKI [142]. Necroptotic cell death induces inflammatory cytokines including, TNF-α, TNF-related weak inducer of apoptosis, and IFN-γ in cisplatin-treated kidneys [142]. Expression of these cytokines further contributes to the induction of receptor-interacting protein 1 (RIP1), RIP3, and MLKL expression in vivo that enhances the necroptotic signaling pathway by positive feedback [142]. Thus, necrotic cell death in renal tubules is dependent of the RIP1/RIP3 and MLKL which is stimulated by cisplatin [142]. Necrosis is caused by high-dose concentrations of cisplatin, whereas apoptosis is stimulated by lower doses [141]. There have been multiple signaling pathways linked to cisplatin-induced apoptosis. These pathways include the intrinsic (mitochondrial), extrinsic, and endoplasmic reticulum stress apoptosis pathways.

#### 4.3.2. Apoptosis-Intrinsic/Mitochondrial Pathway 

Cisplatin has been shown to induce the mitochondrial apoptotic pathway through a variety of physiological processes including ROS production and the release of cytochrome c through stimulation of proapoptotic proteins [89] (Figure 4). Mitochondrial dysfunction is associated with CIAKI [94,148]. A variety of pharmaceutical interventions that have targeted the mitochondrial apoptotic pathway have shown that nephroprotection is linked with a decrease in caspase-3 activity; however, cisplatin increases it [54,64,149]. The antiapoptotic protein Bcl-2 is established to prevent mitochondrial dysfunction-induced apoptosis [150]. Following cisplatin treatment, western blot analysis showed Bcl-2 expression is dose dependently downregulated in HK2 cells, leading to cytochrome c release, caspase-3 cleavage, and apoptosis. Cisplatin-treated cells showed a dose dependent reduction in Sirtuin 5 (Sirt5) expression. Sirt5 overexpression increased bcl-2 expression and reduced apoptosis as determined via Annexin V/PI staining. Results also showed that adjunct Sirt5 and cisplatin therapy reduced bcl-2 expression, a pathway known to induce apoptosis, indicating that Sirt5 exhibits its renoprotection in a bcl-2 independent manner, through activation of caspase-3. Interestingly resveratrol a Sirt5 activator in humans [151] attenuated CIAKI; however, there was no evaluation on Sirt5 expression [152]. It is possible that Sirt5 was activated by resveratrol and inhibited mitochondrial apoptosis by stimulating bcl-2 to inhibit mitochondrial dysfunction. Overexpression of bcl-2 is associated with reduced cisplatin cytotoxicity [153] and therefore may not be a suitable treatment option.

#### 4.3.3. Apoptosis-Extrinsic Pathway 

The extrinsic apoptotic pathway is triggered when a ligand binds to a death receptor located on the cytoplasmic membrane of cells (Figure 4). Activation of the extrinsic pathway contributes to the loss of tubular cells in AKI [156]. The extrinsic apoptotic pathway activates the caspase-8 molecules, which further activates the downstream effector caspase-3 [15,157]. Cisplatin induces the extrinsic pathway through an increase in proinflammatory cytokine TNF-α expression via its death receptor 1 (TNFR1). It is well established in the literature that TNFR1 is a major component of the TNF family, which plays an important role in the extrinsic apoptotic pathway as shown in studies where TNFR1 knockout mice have been suggested to be resistant to CIAKI [157,158,159]. The pathways of apoptosis are inter-related. There are three ways the extrinsic pathway is linked to the mitochondrial pathway that result in caspase-8 activation [160]. ROS generation in the mitochondria results in a direct link to the fas gene and Fas-ligand (Fas-1), which results in apoptosis [157]. In in vitro studies, using NRK-53E cells and α(E)-catenin knockdown cell line (C2) to identify the specific apoptotic pathway stimulated by downregulation of α(E)-catenin (linking protein) [157]. As a result, it was reported that a reduction in a(E)-catenin expression increased the susceptibility of AKI via the Fas-mediated apoptotic pathway, which confirms the role of Fas [157]. ROS generation also results in the phosphorylation of p38 and mitogen-activated protein kinase (MAPK) resulting in increased TNF-α production [161]. Ramesh and Reeves (2005) used a p38-MAPK inhibitor commonly known as SKF-86002 in vivo and reported that TNF-α levels were significantly decreased as well as the inhibition of p38 protected against CIAKI [162]. It was also suggested that a hydroxyl radical scavenger, dimethyl thiourea prevented the activation of the p38-MAPK pathway which completely ameliorated CIAKI [162]. Another study by Wang and colleagues (2018) used DEP domain containing mTOR-interacting protein (DEPTOR) in vivo (in mice) and in vitro with CIAKI [163]. Evidently, DEPTOR expression increased significantly in the kidneys of mice after 3 days of cisplatin treatment [163]. DEPTOR deficiency in both in vivo and in vitro demonstrated protection of the proximal tubular cell apoptosis induced by AKI. This was achieved by inhibition of TNF-α production and p38 MAPK signaling pathway [163]. 

In addition to TNF-α mediated apoptosis, cisplatin activates the p53 pathway, which can be a result of elevated ROS or induced as a response to elevated ROS from the intrinsic pathway. Treatment with cisplatin results in an increased expression and activation of the p53 protein since it regulates apoptosis of proximal tubular cells via activation of repressions of genes that contain promoters p54-binding sites. Activation of p53 occurs because of alteration in the structure of DNA due to cisplatin treatment which causes activation of molecular DNA damage sensors including ataxia telangiectasia and Rad3-related proteins. These proteins further activate checkpoint kinase 2 which phosphorylate and activate p53 [154]. Herein, p53 allows an enhanced accumulation of p53-upregulated modulator of apoptosis (PUMA) in the mitochondria of tubular cells treated with cisplatin. An interaction between PUMA and B-cell lymphoma-extra-large (Bcl-xL) allows Bcl-2-associated X protein (Bax) to permeabilize the mitochondrial membrane, which then subsequently releases cytochrome C causing caspase activation and eventually causing apoptosis of tubular cells [154]. Jiang and colleagues conducted a study using p53-deficient animals and reported a reduction in the induction of PUMA, which protected against cisplatin-induced AKI [154]. 

#### 4.3.4. Endoplasmic Reticulum Stress-Induced Apoptosis

Cisplatin induces cellular stress and via the mechanism can disrupt ER functions, as illustrated in Figure 1 [41]. It has been suggested the ER stress contributes to the disease manifestation of AKI [155,164]. Cisplatin increases the presence of glucose binding protein 78 (GRP78), an indicator of ER stress, in a variety of settings, [10] however, specifically in kidney tissue of both rats and mice [165,166]. GRP78 is a major ER chaperone protein. It is responsible for many processes including the translocation of newly formed polypeptides across the membrane of the ER. It also facilitates the formation of proteins (folding and assembly) and identification of faulty proteins for ER-associated degradation. GRP78 expresses antiapoptotic properties highlighting its function as an ER stress regulator [167]. GRP78 is responsible for keeping ER stress sensors (IRE1, PERK, and ATF6) inactive when a cell is not under stress. The ER stress response is initiated when the accumulation of unfolded proteins triggers dissociation of GRP78 from its receptor, ultimately resulting in the activation of the ER stress sensors [168]. The unfolded protein response reacts to the activation of stress sensors to restore proteostasis through a variety of complex mechanisms. Excessive stress or prolonged restoration of proteostasis triggers the unfolded protein responses apoptosis pathway to override cell survival [155,169]. The precise apoptotic mechanism has been linked to the activation of pro-caspase-12 in renal proximal tubular epithelial cells following cisplatin treatment [170]. Caspase-12 activation occurs in an independent manner of caspase-c function, through caspase-9 triggered effector caspase-3 activation [15]. Multiple pharmacological agents have been used to target ER stress stimulated by cisplatin treatment. Adjunct therapy with cisplatin and the G-protein-coupled receptor 120 receptor agonist TUG891 slightly reduced sCr compared to cisplatin; however, it was still greater than 2 times the concentration of both TUG891 sole treatment and control mice. Interestingly, following cisplatin treatment there was a significant increase in KIM-1 and NGAL gene expression, with TUG891 and cisplatin treatment restoring these levels almost back to control levels. TUG891 and cisplatin treatment also improved tubular injury score indicated by periodic acid Schiff staining. Investigating gene expression of key UPR and apoptotic genes PKR-like endoplasmic reticulum kinase (p-PERK), activating transcription factor 4 (ATF4) and X-box binding protein 1 (XBP1) showed that concurrent treatment with both cisplatin and TUG891 significantly reduced PERK and XBP1, whilst minimally reducing ATF4 compared to cisplatin. The results of this study indicated that UG891 inhibited ER stress and consequently ER-stress-induced apoptosis demonstrated by reduced positive TUNEL staining and reduced caspase-3 expression [171]. This is one of many drugs published to date targeting ER-stress-induced apoptosis in models of CIAKI [41,172,173,174,175,176].

Pharmacological and genetic alterations that specifically target cell death and the various apoptotic pathways to stimulate cytoprotection, although highly effective in producing reno-protective effects, often reduce caspase activation and therefore disrupt or inhibit the cytotoxic pathways of cisplatin. This, although it protects against CIAKI, also diminishes the chemotherapeutic effects of cisplatin in cancer patients, therefore potentially reducing their prognosis. Previous research focused on the assessment of genetic deletion studies investigating the role of key molecules to unearth potential pathways involved in cisplatin-induced AKI. This then led to the magnitude of recent research published this year and the years preceding it, focusing largely on pharmaceutical interventions whereby inhibiting or stimulating key components of cisplatin’s nephrotoxic inflammatory pathways.

## 5. Cisplatin-Induced Acute Kidney Injury: Role of the Immune System 

Inflammatory pathways have been linked to major pathophysiological mechanisms resulting in CIAKI [45,46,52,124,177,178]. Models of AKI resulting from ischemia, sepsis, and nephrotoxicity all were presented with structural and functional changes to the vascular or tubular endothelium. These changes attract immune cells that infiltrate damaged kidney tissue. Extensive research has implicated a large array of cytokines and chemokines into the robust inflammatory response observed in models of CIAKI (Figure 5). 

### 5.1. Toll-Like Receptors and Cisplatin-Induced Acute Kidney Injury

The innate immune system provides constituent primitive first-line defense mechanisms against an extensive repertoire of invading pathogens [179,180]. Integral to the establishment of innate immunity is a distinct class of pattern recognition receptors, referred to as toll-like receptors (TLRs) [181,182]. TLRs are evolutionary-preserved transmembrane type I sentinel glycoproteins, responsible for facilitating host surveillance through the identification of molecular signatures present on pathogens and self/host cells [181,182]. TLRs contain three structural components: (i) an intracellular C-terminal toll/interluikin-1 receptor domain (TIR), (ii) a central helix spanning the plasma or organelle membrane, and (iii) an ectodomain that extends into the extracellular environment or the lumen of the intracellular organelle [183,184]. The cytoplasmic domain is responsible for mediating signal transduction upon association with activating ligands [183,184]. The leucine-rich ectodomain provides diversity and specificity between individual TLRs, as each TLR can respond to different pathogenic (Table 3) and endogenous activating ligands [184,185]. TLR expression is abundant in human non-immune and immune tissues, including cardiac; pulmonary; nervous; hepatic; gastrointestinal; lymphoid; reproductive; and renal [179]. Interestingly, the human genome contains codes for 11 TLRs [186]. However, only 10 functional TLRs are expressed [186]. It has been suggested that the absence of TLR11 may be responsible for human susceptibility to urinary tract infections, as in murine models TLR11 provides resistance against uropathogenic *Escherichia coli* [186]. Activating ligands of TLRs include pattern-associated molecular patterns (PAMP) [182] and danger-associated molecular patterns (DAMP) [187,188]. PAMPs are defined as highly conserved, invariant motifs present on pathogens, that promote microbial survivability [179,189]. DAMPs are host-derived endogenous cytoplasmic or nuclear immunogenic alarmins that are liberated by damaged, stressed, and necrotic cells in the presence or absence of pathogenic infection to restore homeostatic balance [187,188,190]. Independent of the origin of the activating ligand, the resulting end product of sterile inflammation, produced through the myeloid differentiation factor-88 (MyD88)-dependent pathway (TLR1, 2, 4–10) [191] or the TIR-containing adapter-inducing interferon-*β* (TRIF)-dependent pathway (TLR3 and 4) [192], is ubiquitous among TLRs [193]. Activation of TLRs results in the production of inflammatory modulators, including cytokines, chemokines, interferons, and adhesion molecules, promoting the inflammatory response [194].

TLR expression is abundant in non-diseased human renal tissue [179,199,200,201,202,203,204,205,206,207,208,209,210,211,212,213] (Figure 6). Thus, chronic unregulated and unresolving TLR activation may be responsible for immunopathological consequences, as augmented TLR-induced inflammation and increased expression have been reported in a plethora of non-\infectious and autoimmune diseases that target the renal system, including nephrotoxicity [214]; renal disease [56,215,216]; lupus nephritis [202]; and diabetic nephropathy [217]. Furthermore, TLRs have also been implicated as potential drivers of cisplatin-induced pathologies and toxicities, including AKI [55,218,219]; renal injury [220,221,222]; allodynia [218]; and ototoxicity [223]. As demonstrated by TLR-deficient animal models [55,218,219] and polymorphisms in humans [224,225,226,227], the absence or improper function of TLRs may influence susceptibility and severity of pathologies affecting the renal system. Therefore, immunopathological consequences and renoprotective abilities originating from TLR activation in CIAKI are described.

#### 5.1.1. Toll-Like Receptor 2 is Protective in Cisplatin-Induced Acute Kidney Injury

TLR2 is unique, as it requires recruitment of other TLRs to form heterodimers (TLR1 [228]; TLR6 [228]; and TLR10 [229]) to facilitate activation and subsequent signal transduction, promoting inflammation. While it has been postulated that TLR2 can form a homodimer, it has yet to be observed [230]. The ability of TLR2 to form heterodimer complexes with other TLRs enables it to recognize a broad spectrum of pathogens (e.g., bacterium; fungi; helminth; mycobacterium; protozoa; and virus) [231] and host-derived endogenous ligands [232]. Previous studies involving CIAKI have reported a beneficial role of TLR2 [55,233] in the progression of disease development through mediated autophagy, which has been shown to protect renal cells from the detrimental effects of cisplatin-mediated cell death [234]. Exacerbated CIAKI has been reported in mice with TLR2 deficiency, receiving daily intraperitoneal injections of cisplatin (20 mg/kg/day) [233]. Within 24 h of receiving cisplatin, TLR2-deficient mice displayed significantly increased levels of sCr and BUN, and severe morphological changes to renal tissue, including loss of brush-border cells, tubule dilation, and cast formation [233]. Furthermore, TLR2-deficient mice had reduced renal tubular epithelial cell autophagy, associated with increased protein levels of p62 (an autophagy substrate chaperone, involved in delivering of ubiquitinated peptides to autophagosomes for proteasomal degradation [235]), and decreased levels of microtubule-associated protein 1A/1B light chain 3 II (LC3 II) (a LC3-phosphatidylethanolamine conjugate that is integrated into lipid membranes of autophagosomes and phagophores, which interacts with p62 to select targets for degradation [236,237]) and phosphorylation of phosphoinositide 3-kinase and protein kinase B [233]. This study also reported significantly increased mRNA and protein expression of TLR2 in wild-type mice 24 h after cisplatin treatment, which continued to increase in a time-dependent manner [233]. A complimentary study also reported a protective role of TLR2 during CIAKI, as the absence of TLR2, in mice treated with cisplatin (20 mg/kg), resulted in renal dysfunction shown by increased levels of sCr, BUN, and urea [55]. Histological assessment determined that mice with TLR2 deficiency developed exacerbated renal injury and structural damage, including enlargement of glomerular cavity, renal tubular dilation, renal epithelial cell detachment, formation of renal casts, and infiltration of inflammatory cells (macrophages and neutrophils) into the renal medulla and presence of necrosis [55]. Additionally, lack of TLR2 reduced survivability and failed to protect mice from cisplatin-induced weigh loss [55]. This study also demonstrated a protective role in renal autophagy during CIAKI through induction of the TLR2 pathway, as autophagosome forming molecules may depend on the TLR2 signal [55]. Renal tubular cells isolated from TLR2-deficient mice had decreased levels of autophagy genes (LC3 and autophagy-related 5) and proteins (LC3II, autophagy-related protein-5 and CCAAT-enhancer-binding protein-homologous protein) after cisplatin treatment [55]. Taken together, these results suggest that TLR2 plays a beneficial role during CIAKI by regulating autophagy and reducing renal dysfunction and pathological changes.

#### 5.1.2. Toll-Like Receptor 4 Has a Detrimental Role in Cisplatin-Induced Acute Kidney Injury

Since its discovery in 1997 [238], TLR4 has been extensively investigated in the literature, and its structure, function, and signal transduction remain the most characterised and established of the TLRs [239]. TLR4 plays an essential role in host Gram-negative immunity by identifying lipopolysaccharides, a lipid and polysaccharide conjugate that is a major component in the outer membrane of Gram-negative bacteria [240]. However, the ability of TLR4 to recognize a repertoire of pathogens has been reported, including enveloped viruses [241] and viral proteins (viral fusion proteins [242] and glycoproteins [243]), Gram-positive bacteria [244], and helminths [245]. Furthermore, TLR4 can respond to a broad range of DAMPs, including endoplasmin [239], high-mobility group-1 [246], and heat-shock protein 70 [247], which have been shown to be upregulated during cisplatin treatment. Interestingly, TLR4 has also been shown to participate in transition metal sensing [248], and metals, including nickel, cobalt, and platinum [223,248] have been observed as TLR4-activating ligands. Due to the increase in circulating DAMPs and cisplatin (a platinum-based chemotherapy) acting as TLR4 ligands, TLR4 has been implemented as a major contributor in driving pathogenesis and development of CIAKI through upregulation of inflammation and proinflammatory and subsequent renal dysfunction, renal tissue injury, and nephrotoxicity [239]. A study involving TLR4-deficient mice administered a toxic dose of cisplatin (20 mg/kg) to induce acute renal failure within 72 h and reported significantly reduced markers of inflammation, nephrotoxicity, and renal function and decreased renal injury and histological abnormalities [239]. To determine the effect that TLR4 deficiency has on cisplatin-induced renal dysfunction and structural changes, BUN and sCr were used as indicators of function [239]. Severe renal failure (indicated by elevated levels of BUN and sCr) was observed between 48 and 72 h in WT after bolus dose of cisplatin and was accompanied by histological abnormalities including advanced tubular injury, cast formation, absence of brush-border membranes, shedding of tubular epithelial cells, necrosis of renal tubule cells, and dilation in renal tubules [239]. Mice with TLR4 deficiency has significantly preserved renal function 72 h after administration to cisplatin, as shown by reduced BUN ad sCr concentrations and minimal histological changes. Thus, indicating that TLR4 contributes to structural and functional consequences during CIAKI [239]. Furthermore, immunopathological consequences, including reduced leukocyte infiltration, decreased concentrations of cytokine and chemokine in serum (i.e., TNF-*α*; IL-1*β*; IL-2; IL-6; and IL-10), kidney (i.e., TNF-*α*; IL-6; CCL5; MCP-10; and KC) urine (i.e., TNF-*α*; IL-2; IL-6; CCL5; MCP-1; KC; and IP-10), and reduced activity of p38 MAPK and JNK phosphorylation was also observed in TLR4-deficient mice when compared to WT [239]. Thus, suggesting that the potent inflammatory response initiated by TLR4 may be responsible for initiating CIAKI [239]. Therefore, a tailored therapy encompassing a combination of cisplatin and a TLR4 inhibitor is an appealing approach to preserve renal structural integrity and preservation of function in CIAKI.

A potential TLR4 inhibitor to be used in conjunction with cisplatin, which has shown promising results in septic-induced AKI, is resatorvid (TAK242) [249,250]. TAK242 is a cyclohexene derivative [251], which exerts its inhibitory effect by binding to the intracellular domain of TLR4. Upon binding, TAK242 causes a confirmation change in the cytoplasmic tail, which results in the inability of the bridging adaptor molecules TIR-containing adapter protein/myeloid differentiation factor-88 and translocating chain-associated membrane protein/TRIF to associate [252], thus preventing TLR4 signal transduction and subsequent production and release of proinflammatory mediators [252]. Administration of TAK242 to ovine models of Gram-negative bacteria resulted in enhanced renal function, demonstrated by abolishment of impaired Cr clearance in the urine, reduced BUN and sCr, prevention of renal hypoperfusion, and reduced swelling of endothelial cells in glomerular capillaries [249,250]. Additionally, a recent article has shown that TAK242 is able to enhance the cytotoxic effect of cisplatin in breast and ovarian cancer cells, while preventing its toxic effects of cells [253]. Thus, suggesting that dual treatment with TAK242 and cisplatin could enable a reduced dose of cisplatin given to patients. Taken together, TAK242 should be further investigated in CIAKI, as it represents a pharmaceutical that could (i) prevent detrimental chronic inflammation, (ii) retain structural integrity and renal function, and (c) intensify the effect of cisplatin in CIAKI.

#### 5.1.3. Toll-Like Receptor 9 is Protective in Cisplatin-Induced Acute Kidney Injury

TLR9 is a cytosolic receptor bound to the membranes of intracellular organelles (e.g., endosomes, lysosomes, and endolysosomes) and is responsible for identification of unmethylated cytosine-phosphate-guanosine DNA present in bacteria and viruses [211,254,255]. Furthermore, TLR9 has been shown to directly contribute to cardiac, hepatic, and renal ischemic tissue injury [211] and lupus nephritis [256,257] through recognition of endogenous mitochondrial DNA products [258], chromatin IgG complex [259], GP96 [260], high-mobility group box-1 [261], and heat-shock protein 90 [262]. A study involving CIAKI determined a protective immunomodulating role of TLR9 in disease progression, as TLR9 absence resulted in accelerated and increased pathological development [56]. TLR9-deficient mice were administered cisplatin (20–25 mg/kg) via intraperitoneal injection and were then euthanised 24–72 h after treatment [56]. When compared to wild-type controls, TLR9-deficient mice had greater renal dysfunction and histological injury 24 h after cisplatin exposure, as shown by significant increases in serum urea, tubular injury score, neutrophil, and CD4+ cells and mRNA expression of CXCL1/2 [56]. Additionally, augmented disruption of renal tubular structure and integrity, tubular necrosis, cast formation, and accumulation of tubular cell debris was observed in kidneys derived from TLR9-deficient mice when compared to wild-type controls [56]. However, after 56 h of cisplatin exposure, no significance in renal function and histological injury was observed between TLR9-deficient mice and wild-type controls, suggesting that increased renal injury was not mediated by effector cell function in TLR9-deficient mice [56]. Previous in vitro studies involving human cells have postulated simultaneous activation of regulatory T cells (Tregs) and TLR9, suggesting the ability of TLR9 activation to regulate/suppress Treg activity [263]. Therefore, using this hypothesis, the authors determined if exacerbated renal injury and dysfunction was caused by Treg cell activity modulated by TLR9 [56]. The findings from this study showed that TLR9-deficient Tregs were not defective in functionally, abundance or apoptotic-inducing abilities but that the amount of adhesion molecules responsible for Treg recruitment into the kidneys was reduced in TLR9 absence [56]. This is supported by literature that shows that Tregs must be actively recruited into renal tissue to exert renoprotective abilities in AKI [56,264]. Therefore, taken together, this study suggests that TLR9 plays a beneficial role in CIAKI by enabling recruitment of Tregs into inflamed renal tissue [56].

### 5.2. Cytokines 

#### 5.2.1. Tumor Necrosis Factor Alpha

Cisplatin-induced nephrotoxicity involves the activation of a proinflammatory response [265]. Cisplatin has been linked to the increased expression of TNF-α in both serum and urine concentrations of AKI [31]. TNF-α is a branch of the TNF family. These proteins are important cytokines responsible for cell signaling. They play an important role in immunity as well as the possession of proinflammatory properties [15,266]. TNF-α inhibitors have shown that in the absence of TNF-α.

α cisplatin-induced nephrotoxicity is attenuated, indicating that TNF-α plays a significant role in cisplatin’s nephrotoxic effects. Treatment with the matrix metalloprotease inhibitor GM-6001, a TNF-α antagonist, reduced urea levels in GM-6001 mice compared to cisplatin, showing treatment with GM-6001 enhanced renal function. This improvement in renal function correlates with the improved renal histology [33]. Genetic modification of TNF-α also displayed renoprotective effects. Silencing of TNF-alpha showed amelioration of kidney dysfunction resulting from cisplatin treatment, with TNF-alpha −/− mice protected against the nephrotoxic effects of cisplatin [33]. This highlights a crucial role of proinflammatory cytokines and chemokines in the pathogenesis of CIAKI, particularly a central role for TNF-alpha [33]. Interestingly, TNF-α enhances the anticancer properties of cisplatin in breast cancer cells, both in vitro and in vivo [267]. It would be interesting to investigate the effects of TNF-α downregulation on the cytotoxicity of cisplatin in an in vivo model of CI-AK. In addition to upregulation by cisplatin, Nfκb is also activated by TNF-α [268] and in CIAKI [33]. 

#### 5.2.2. Nuclear Factor Kappa-Light-Chain Enhancer of Activated B Cells 

Nuclear factor kappa-light-chain enhancer of activated B cells (NF-κB) is a key prosurvival and inflammatory cell transcription factor and is induced by cisplatin treatment [269]. Expression of Nfκb was significantly upregulated in rats treated with cisplatin compared to control rats. SA pretreatment significantly downregulated NF-kB expression compared to cisplatin [103]. Interleukin-6 (IL-6) and TNF-α levels were markedly reduced in the SA and cisplatin compared to only cisplatin-treated rats; however, levels were not completely restored to control levels. Quercetin is a flavonoid that elicits anticancer, anti-inflammatory, and antioxidant properties. Results of this study showed quercetin restored kidney function indicated by reduced serum BUN and creatinine compared to cisplatin-treated mice. Quercetin also downregulated mRNA expression of key inflammatory markers IL-1β, IL-6, and TNF-α compared to control, indicated by real-time PCR. Western blot analysis indicated Quercetin treatment decreased Syk/NF-κB activity in kidney tissue following cisplatin-induced AKI. They concluded that inhibition of Mincle/Syk/NF-κB signaling by quercetin exerts its renoprotective effects through reduced inflammation [46]. The progression would be to test quercetin in a tumor bearing model of cisplatin-induced AKI, and it is one of the first anticancer drugs to be used and could potentially exert its renoprotective effects whilst maintaining cisplatin’s cytotoxicity. This is a promising therapeutic option for cancer patients. In addition to pharmacological inhibition of NF-kB displaying renoprotective effects, transcriptional inhibition of NF-kB also protects against cisplatin-induced AKI [269]. The literature has investigated the role of transforming growth factor-β-activated kinase-1 (TAK1) in cisplatin-induced acute kidney injury [54,270]. TAK-1 is a component of the NF-kB pathway and protects cells from TNF-α-induced cell death through upregulation of antiapoptotic proteins [271]. TAK1 expression is known to be upregulated in response to cisplatin treatment [270]. TAK1 gene disruption was performed in order to generate a knockout in the proximal tubule to investigate the role of TAK1 following cisplatin treatment. Lower sCr and BUN and levels were observed in TAK1-deficient mice 72 h following injections of 20 mg/kg of cisplatin compared to controls. This is indicative of reduced renal dysfunction in the TAK-1-deficient group. H&E staining of kidney sections investigating tubular epithelial cell injury, proximal tubular dilation, and cast formation showed TAK-1 deficiency reduced histological evidence of kidney injury [54]. Following this knockout study, a pharmacological study of cisplatin-induced AKI was performed using a TAK1 inhibitor. Results of the study determined that TAK-1 inhibition increased sCr and BUN, suggesting TAK-1 plays a protective role against CIAKI. Interestingly, following the increased expression of TNF-alpha and consequently Nfκb observed in CIAKI, this is often correlated with a downregulation in Interleukin-10 (IL-10).

#### 5.2.3. Interleukin-10 

There is significant evidence implicating proinflammatory cytokines in the pathogenesis of cisplatin-induced renal dysfunction [272,273,274], stimulating the focus on anti-inflammatory cytokines as therapeutic targets against CIAKI. IL-10 is an anti-inflammatory cytokine produced by T helper cells, T cells, dendritic cells, and macrophages [31]. IL-10 inhibits many physiological processes including early-phase inflammation, cytokines, chemokines, neutrophil activation, and NO production [273]. Deng and colleagues conducted a study to investigate the effects of IL-10 on CIAKI in male BALB/c and C57BL/6 mice. A histological analysis was conducted 72 h after mice were treated with cisplatin. Results showed evidence of necrosis in the renal proximal tubule and the straight tubule. Cast formation and leukocyte accumulation was also shown, indicative of an inflammatory response. The results of the study showed that serum concentrations of IL-10 were downregulated in response to cisplatin treatment and maximal inhibition of renal injury was observed when cisplatin and 1g of IL-10 (determined via dose-dependent study) was administered concurrently. It was also determined that IL-10 administration one hour post cisplatin treatment also decreased renal damage [273]. A separate study in IL-10 KO mice showed enhanced CIAKI, further explicating its renoprotective effects [275]. Increased IL-10 expression has also been associated with the renoprotective effects of AT_2_ receptor stimulation following TLR4-induced inflammation [118]. This shows that enhanced expression of IL-10 is renoprotective and finding a pharmaceutical activator of IL-10 to assess in a model of AKI may provide therapeutic benefit. Tumor cells are known to overexpress key cytokines, such as IL-10. IL-10 protected against chemotherapeutic agent effects, through upregulation of antiapoptotic proteins such as Bcl-2 and Bcl-xL [276]. As such, further investigations into the effects of IL-10 attenuation of CIAKI in a tumor-bearing model should be completed. IL-33 another cytokine has been suggested to contribute to cisplatin’s nephrotoxicity.

#### 5.2.4. Interleukin-33

IL-33 is a proinflammatory cytokine. This cytokine contains a receptor ST2, which attracts immune cells such as CD4+ and T cells via chemotaxis [31]. Elevated kidney expression of IL-33 was observed in CIAKI and tubular injury, suggesting that IL-33 may contribute to the nephrotoxic effects detected following cisplatin treatment [50,277]. Akcay and colleagues conducted a study using a decoy receptor to identify the role IL-33 and CD4+ T cells play in cisplatin-induced acute tubular necrosis (ATN) and apoptosis. In the study, two experimental groups were used. The first group was administered sST2, a decoy receptor to inhibit IL-33 function. It was found that when this was administered, a reduction in the infiltration of CD4+ T cells was observed along with decreased ATN and apoptosis. In the second group, recombinant IL-33 was administered and was found to intensify CIAKI, whilst in CD4+-deficient mice administered with IL-33, kidney structure and function remained unaffected. The results of this inferred that CD4+ T cells were accountable for the damage caused by IL-33; in addition they concluded that IL-33 inhibition may have a therapeutic potential in CIAKI [277]. Ravichandran and colleagues conducted a study that investigated the protective role of IL-33 deficiency on cisplatin-induced AKI. The results showed that IL-33 deficiency was not protective against CIAKI in mice [50]. The interesting component of this study that differs from most CIAKI models is that it was a tumor-bearing model. This stimulated investigations regarding not only the nephroprotective effects of IL-33 deficiency but also the effects of the deficiency on cisplatin’s anticancer activity. BUN and sCr were elevated in both IL-33 deficiency and WT mice treated with cisplatin in addition to increased expression of the AKI biomarker NGAL. ATN and tubular apoptosis observed in cisplatin-induced AKI were not alleviated in IL-33-deficient mice, furthering the evidence that IL-33 deficiency could not attenuate cisplatin’s nephrotoxic effects. It has been demonstrated that IL-33 is upregulated in cisplatin treatment [277]. Although IL-33 may be involved and contribute to CIAKI, it is unlikely to be causative. To investigate IL-33 deficiency on the cytotoxicity of cisplatin, cleaved, caspase-3 expression was examined. In tumors of vehicle, WT-treated and IL-33-deficient mice, no significant differences in caspase-3 expression was observed; however, caspase-3 levels were elevated in tumors of cisplatin WT mice but not in IL-33-deficient mice. Although there was evidence of reduced apoptosis, tumor weight, tumor volume, and tumor growth were all reduced in IL-33-deficient mice, a beneficial outcome for cancer prognosis [50]. Recent publications have implicated tumors themselves in the pathogenesis of AKI potentially through tumor lysis syndrome (TLS). TLS is caused by cancer treatments; however, they can occur spontaneously themselves, discharging cancer contents into the bloodstream, clinically presenting as hyperuricemia, hyperkalemia, hyperphosphatemia, and hypocalcemia which can manifest and result in renal dysfunction [278]. Most publications to date use noncancer-bearing models to assess interventions for CIAKI. Healthy mice are injected with cisplatin either prior, concurrently, or following the intervention. This therefore investigates nephrotoxicity caused by cisplatin itself; however, it excludes the influence of cancer on kidney dysfunction. It has been suggested that TLS from cancer patients may contribute to the pathogenesis of CIAKI; therefore, Ravichandran and colleagues repeated the same experiments on IL-33 deficiency in cisplatin-treated mice without cancer. The results of this concluded that cancer did not influence the lack of renoprotection, elicited by IL-33 deficiency [50]. Knowing this, it highlights the importance of trialing therapies in both tumor-bearing and tumor-absent models of CIAKI. 

#### 5.2.5. Interleukin-6 

IL-6 is a pleiotropic cytokine predominantly exerting proinflammatory functions but also exhibits anti-inflammatory properties [279]. It is a member of the interleukin family of cytokines, a collective whose involvement has been extensively researched in the pathology of CIAKI for many years. Serum and urine levels of IL-6 are elevated following cisplatin treatment, and it has been used as an early serum and urine biomarker of AKI [280]. However, IL-6 stimulation has been proposed to play a protective role in models of cisplatin-induced AKI implied to be via upregulation of antioxidant markers [49]. IL-6 −/− mice were administered 30 mg/kg of cisplatin via intraperitoneal injection. Twenty-four and 72 h post cisplatin administration blood and kidneys were harvested and analyzed for oxidative and antioxidative stress markers. Western blotting of IL-6 −/− cisplatin-treated mice showed increased expression of 4-HNE compared to wild-type mice. Following this, gene expression of free radical scavengers SOD1 and SOD2 in cisplatin-treated kidneys was analyzed via RT-PCR assays. Kidneys excised 24 h after cisplatin treatment yielded no significant difference in SOD1 expression; however, SOD1 expression was reduced in wild-type and IL-6 −/− mice 72 h after cisplatin treatment. In addition to investigating gene expression, enzymatic activity of SOD was analyzed indicating that SOD activity was significantly reduced in kidneys of IL-6 −/− mice compared to wild type [49]. Previously, it was determined that although IL-6 deficiency did not accelerate the development of systemic injury, it did accelerate progression of cisplatin-induced acute renal failure [281]. This indicates that IL-6 may play a protective role in CIAKI, but to date, there is no pharmacological IL-6 antagonism studies investigating this. Cancer models have showed that increased expression of IL-6 enhances acquired cisplatin resistance and reduces cytotoxicity; therefore, direct stimulation of IL-6 over-expression may reduce the efficacy of cisplatin [282] and therefore may not be an appropriate therapeutic target for cancer patients to prevent CIAKI. Many cytokines have been explored in models of CIAKI; however, recent literature has elucidated the role of specific chemokines in the pathogenesis of the disease. 

### 5.3. Chemokines 

The development and progression of an inflammatory response has long been associated with kidney damage and nephrotoxicity stimulated by cisplatin treatment [283]. Evidence has presented a role for proinflammatory chemokines as potential mechanisms associated with CIAKI, which is suggested to be mediated via TNF-α [33]. Both CXC and CC chemokines have varying roles in cancer, with many members from both subfamilies recruited to tumor sites playing either a pro- or anti-tumor role. Multiple chemokines from the CXC family have been linked to the pathogenesis of cancer, particularly in the stimulation of angiogenesis [284]. Several chemokine family members have also been shown to contribute to the pathogenesis of CIAKI. 

#### 5.3.1. CXCL16 

CXC chemokine ligand 16 (CXCL16) is a member of the CXC family of chemokines [47]. CXCL16 has been associated with proinflammatory properties in multiple diseases [285,286]. Research suggests that CXCL16 may play a role in acute coronary syndrome most often associated with atherosclerosis [287]. Inhibition of CXCL16 has been linked to ameliorating effects in a variety of inflammatory related diseases such as liver inflammation and steatohepatitis [288], anti-GBM glomerulonephritis [289] and the regulation of CIAKI [47]. CXCL16 expression is upregulated in kidneys following cisplatin treatment [47]. To further understand the mechanisms stimulating this upregulation and the role CXCL16 plays in inflammation and apoptosis in renal tubular cells, wild-type and knockout mice were injected with 20 mg/kg of cisplatin. Seventy-two h after cisplatin treatment, the mice were culled. The results showed that kidney dysfunction was observed in wild-type mice, indicated by elevated levels of serum BUN. CXCL16 knockout mice were protected from cisplatin-induced renal dysfunction with reduced levels of serum BUN. Knockout mice also presented with reduced histological damage of renal tissue compared to WT mice. This study used RT-PCR to investigate mRNA expression of key proinflammatory cytokines associated with CIAKI. Results showed that following cisplatin injection expression of TNF-a, IL-1β, IL-6, and transforming growth factor β1 were upregulated in wild-type mice compared to the vehicle. Inhibition of mRNA expression of these molecules was observed in CXCL16 knockout mice. Decreased caspase-3 activation was also seen in CXCL16 −/− mice illustrating CXCL16’s role in tubular epithelial cell apoptosis. [47]. A separate study also involving CXCL16 −/− mice investigated renal injury associated with salt-sensitive hypertension. The study showed that DOCA-salt-treated −/− mice showed reduced renal dysfunction, proteinuria, BUN, and fibrosis compared to wild-type mice [290]. The collection of these studies has highlighted a role for CXCL16 in the inflammatory response associated with renal injury. Given the information and results observed in these studies, the progression would be to assess concurrent treatment with cisplatin and the pharmacological inhibition of CXCL16 in the prevention of nephrotoxicity. Pharmacological treatment using a monoclonal, rat anti-mouse CXCL16 neutralising antibody has shown to reduce liver inflammation in chronic hepatic injury [291]. More relative to AKI, a study involving the pharmacological inhibition of CXCL16 using an antiserum generated against CXCL16 showed reduced progression of anti-GBM glomerulonephritis suggested to be through its role in leukocyte influx [289]. Despite the evidence linking CXCL16 to CIAKI, it is not the only CXC chemokine linked to the disease. The CXCL1-CXCR2 axis has also been shown to be involved in renal damage following cisplatin treatment.

#### 5.3.2. CXCL1-CXCR2 Axis

The chemokine (C-X-C motif) ligand 1 (CXCL1) belongs to the CXC family of chemokines. The molecular structure of CXCL1 can be either a monomer or a dimer and is a highly potent CXCR2 receptor agonist [292]. The CXCL1-CXCR2 axis has been implicated in a variety of inflammatory diseases, specifically through its role in neutrophil recruitment and microbial death at sites of tissue injury [292,293]. A key component in the immune response triggered by AKI promoting renal injury is the accumulation of key leukocytes, such as neutrophils and monocyte/macrophages [294]. CXCL1 and its upregulation has been secondarily investigated in models of CIAKI specifically in IL-33-deficient mice [50]. A recent study published demonstrated the inhibition of the CXCL_1_-CXCR_2_ axis provided ameliorating effects against renal damage induced by CIAKI [35]. Following cisplatin treatment, a significant increase in kidney mRNA and protein expression for both CXCL1 and CXCR2 compared to control was observed, indicating that the CXCL_1_-CXCR_2_ axis plays a role in the nephrotoxic effects induced by cisplatin. Reduced sCr and BUN were observed in both CXCL1- and CXCR2-deficient mice, indicating axis silencing improves renal function and AKI induced by cisplatin. There was also evidence of reduced renal neutrophil infiltration, indicating the potential of a reduced immune response in both CXCL1- and CXCR2-deficient mice. Following the genetic deletion studies, pharmacological inhibition of CXCL1 and CXCR2 with repertaxin displayed ameliorating effects against CIAKI. The results of their study concluded that inhibition of the CXCL_1_-CXCR_2_ axis was renoprotective against CIAKI through inhibition of p38 and Nfκb. [35]. Following this, the use of selective pharmacological antagonists of CXCL1 to and CXCR2 is needed to observe their isolated role in the disease pathogenesis of CIAKI.

## 6. Preclinic to Clinic Translation 

Despite the magnitude of preclinical studies aimed at preventing CIAKI/nephrotoxicity, only a small number have progressed to preclinical stages. Interestingly, most models of CIAKI begin in a nontumor bearing model, with healthy mice that are treated with either a single high dose of cisplatin or multiple lower doses over an extended period. Mice present an important preclinical model in research, with a 99% similar genome to humans. Additionally, their small size provides a cost-effective alternative for large-scale studies. Many drugs have been validated in mice for efficacy and safe dosage, which have safely translated to humans [295]. However, there are also instances where this translatability has been questionable. In 2019, Leena and colleagues published a paper presenting an overview of the animal-to-human translational success rates, where they concluded that translational success is unpredictable. They also pointed out that the data used were old and potentially biased and suggested that the data from newer papers were unreliable [296]. This was a paper published recently and it is interesting the lack of success rates published by newer studies. They do also suggest that suboptimal experimental design may be a contributing factor to translational failure in recent models. Reproducibility in both animal and human studies has also posed an issue regarding translatability [296]. In 2015, a paper was published specifically addressing five key improvement areas for preclinical models to improve drug development for CIAKI. Animal-based experiments are often the basis for clinical trial studies. A perfect example is the studies investigating the effects of *N-*acetylcysteine (NAC) in the prevention of contrast-induced AKI. In preclinical models, NAC data were supportive of its renoprotective effects warranting its use in clinical trials; however, there were key elements missing from the preclinical studies. Preclinically NAC was administered via IV, whilst in the clinical trial, it is was received orally, and there were no data obtained in the preclinical findings indicating half-life or duration of antioxidant effects and that could explain the failure in translation [297]. 

As mentioned earlier, cancer itself can cause renal damage through the biproducts released in cancer cell degradation stimulated by cancer itself or its treatment [298]. As such, it is critical that tumor-bearing models are tested following nontumor-bearing studies, given clinical trials will always be in patients with cancer. Majority of preclinical studies adapt a single high-dose injection with endpoint being 72 h after cisplatin, it may also be beneficial to complete a model in addition to this that closer replicates human cisplatin drug regimens before moving into clinical trials. It is also interesting that even despite all the preclinical drugs published that demonstrate amelioration of CIAKI, there is no evidence of the drug progressing further into a tumor-bearing model or clinical trials. A paper published in 2019 presented a drug that can prevent CIAKI whilst increasing cisplatin cytotoxicity as demonstrated in H1299 cancer cell lines [37]. This presents highly promising potential and hopefully a follow-up study by the same group in a tumor-bearing model will be a follow-up before its progression to clinical trials.

## 7. CIAKI Clinical Trials 

There have only been a few clinical trials to date as described in Table 4. Not all clinical trials were completed or present results. Clinically Amifostine is the only FDA approved treatment in the prevention of CIAKI. It has undergone a variety of clinical trials in cancer patients treated with cisplatin, with result demonstrating its renoprotection and cytoprotection. However, its cytoprotection has yielded inconsistent results with toxicities still occurring despite the use of optimal Amifostine doses [299]. Another clinical study which is continuously disagreed upon throughout the literature is the safety and efficacy of mannitol as a therapeutic option for CIAKI [300,301]. Mannitol is an osmotic diuretic which has generated promising preclinical in vivo data in the prevention of cisplatin-induced acute kidney injury [302]. These preclinical data then provided the evidence required to progress the use of mannitol into clinical trials, leading it to FDA approval and the current standard of care for patients with CIAKI. However, there is evidence of increased hyponatremia in cisplatin-treated patients hydrated with saline and mannitol [303]. In addition to mannitol, another more recent clinical trial prevention strategy for CIAKI is magnesium preloading. Results of this study showed that although magnesium preloading prevented cisplatin-induced acute kidney disease, there was no statistical significance in the prevention of CIAKI. However, there was a reduction trend, and thus a larger-scale trial is required to further assess its amelioration of CIAKI [304]. Pretreatment with pantoprazole has provided an interesting therapy, providing ameliorating effects in preclinical findings [62]; however, in the clinical setting, results showed pantoprazole unprotective preventing cisplatin nephrotoxicity in osteosarcoma patients. This provided further evidence for cancer cell influence on nephrotoxicity and the importance of preclinical tumor-bearing models and publication of the findings of these studies. A universal issue that is observed throughout most clinical trials of CIAKI is a lack of statistical significance occurring due to small sample size. This small sample size is largely attributed to the harsh exclusion criteria for this disease; generating a larger sample size would benefit clinical trials for CIAKI therapeutic studies. 

## 8. Potential Future Treatments 

The current preventative treatments for cisplatin-induced nephrotoxicity are hyper-hydration with intravenous saline, sodium loading (shown to have no influence on cisplatin-induced nephrotoxicity incidence [310]) and forced diuresis with mannitol [301,311]. Mannitol treatment has yielded conflicting results [312]. Mannitol has shown however, to cause over diuresis resulting in dehydration [95]. Physiologically cisplatin is known to reduce medullary blood flow and induce afferent arteriole vasoconstriction causing ischemic damage [31,313]. Normal autoregulatory renal vasodilation occurring in response to ischemia is instead replaced with enhanced vasoconstriction resulting in further hypoxic kidney injury [31]. We therefore hypothesize that in addition to hydration, treatment with a vasodilatory inducing or enhancing drug may work to flush cisplatin out of the kidney system. The use of drugs capable of stimulating vasodilation through other pathways that also promote cytotoxicity of cisplatin is another proposal. Currently a large proportion of successful preclinical therapies target antioxidant and anti-inflammatory pathways, which consequentially also result in antiapoptotic mechanisms. This poses an issue when cisplatin is given to promote cancer cell death to improve the prognosis for cancer patients. Stimulation of antiapoptotic mechanisms interfere with the anticancer pathway of cisplatin which makes giving a therapy to reduce cisplatin cytotoxicity to promote renoprotection counterproductive. Based on this, we propose the use of certain drugs capable of stimulating vascular relaxation to promote excretion of cisplatin using drugs that also enhance the cytotoxicity of cisplatin. TIC10, a chemotherapy currently in clinical trials as an anticancer agent is a TRAIL-inducing compound. TRAIL has been linked to potential of relaxation enhancement through increased eNOS shown in HUVECs. Our laboratory also published a paper showing the enhancement of vascular relaxation of healthy vessels by TIC10 [314]. Interestingly, given the anticancer properties of TIC10, more specifically its ability to selectively induce apoptosis in cancer cells whilst sparing healthy cells in addition to its relaxation properties, it presents a possible therapeutic option in the prevention of CIAKI.

## 9. Conclusions

Despite advances in oncology research and interventions, research is yet to yield a solid and universal solution to the detrimental nephrotoxic side effects experienced by patients treated with cisplatin [315]. To date, many pharmacological interventions have proven effective in the prevention of CIAKI. However, their nephroprotective effects directly interfere with the cytotoxic pathways of cisplatin and thus reduce its efficacy. Several potential therapies have been discovered that prove beneficial in the reduction of AKI in a preclinical research setting. However, the transference from animal to human clinical studies is disappointing [297]. This warrants the need for research into uncovering additional pharmacological interventions to develop a more translatable treatment. There is an extensive pathophysiological map identifying mechanisms involved to further develop understanding into this disease and how it manifests to create better prevention or treatment options. The purpose of this review is to present a large collective of recent studies as well as the most influential past studies to detail the extent of the complexity of this disease. It also elucidates the potential issues arising that are preventing preclinic to clinic translatability. Currently, despite the hundreds of preclinical promising models, there is a sever lack of progression of these therapies to tumor-bearing models and further progression to clinical trials. Many promising therapies also directly target cisplatin cytotoxic pathways and are therefore not feasible to progress with into clinic. Cellular uptake/efflux, oxidative stress, vascular injury, necrosis, apoptosis, and inflammation remain the focused therapeutic areas; however, perhaps this disease requires a broader treatment or a regimen of treatments to combat the array of mechanisms. 

## Figures and Tables

**Figure 1 cancers-13-01572-f001:**
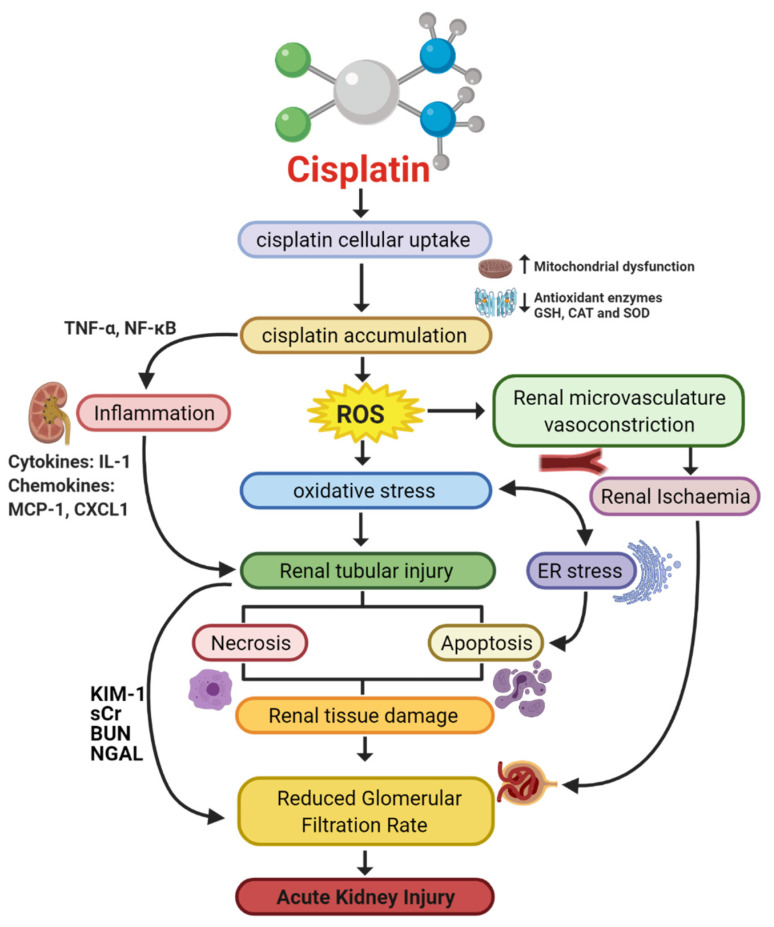
Pathophysiological map of the key molecular pathways demonstrated to play a role in the pathogenesis of cisplatin-induced acute kidney injury (AKI). The mechanisms associated with cisplatin-induced AKI (CIAKI) are complex, and the relationship between the key pathways remains unknown. However, it is believed that the detrimental nephrotoxic effect of cisplatin in renal tissue is due to platinum accumulation. Cisplatin accumulation triggers increased production of tumor necrosis factor alpha (TNF-α) [33,34] and reactive oxygen species (ROS), stimulating inflammation [35], oxidative stress [36], vascular injury [31], and apoptotic pathways [37]. The apoptotic mechanisms then promote renal tissue damage leading to the key clinical manifestation of nephrotoxicity (a reduction in glomerular filtration rate (GFR)) resulting in CIAKI. Abbreviations: GSH, glutathione; CAT, catalase; SOD, superoxide dismutase; TNF-α, tumor necrosis factor alpha; ROS, reactive oxygen species; ER stress, endoplasmic reticulum stress; and GFR, glomerular filtration rate. IL-1, Interleukin 1; MCP-1, monocyte chemoattractant protein 1; CXCL1, C-X-C Motif Chemokine Ligand 1; KIM-1, Kidney Injury Molecule 1; sCr, Serum Creatinine; BUN, Blood Urea Nitrogen; NGAL, Neutrophil gelatinase-associated lipocalin. Figure adapted from “Cisplatin nephrotoxicity: mechanisms and renoprotective strategies” by N. Pabla and Z. Dong, 2008, Kidney International, Volume 73, P994-1007, Copyright [2008] by the Elsevier.

**Figure 2 cancers-13-01572-f002:**
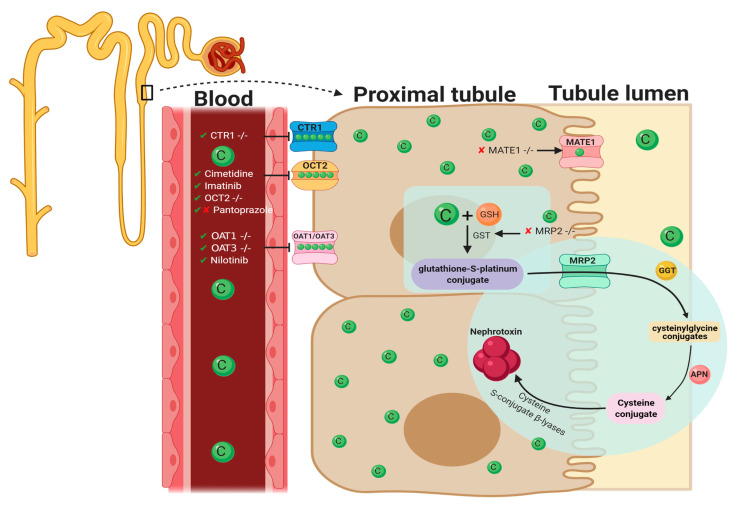
Graphical representation of key molecules and pathways involved in cisplatin transportation initiating nephrotoxic effects. Key transporters responsible for cellular uptake of cisplatin from the blood into PTECs resulting in a much greater platinum concentration compared to the blood. The key interventions trialed to date and their effectiveness in targeting CIAKI are also illustrated. Diagram details the cellular processes involved in the cellular uptake [57,58,59,60,61,62,63,64], efflux [65,66], and metabolism of cisplatin into a highly reactive thiol (nephrotoxin) [67] and the treatment targeted to prevent them.

**Figure 3 cancers-13-01572-f003:**
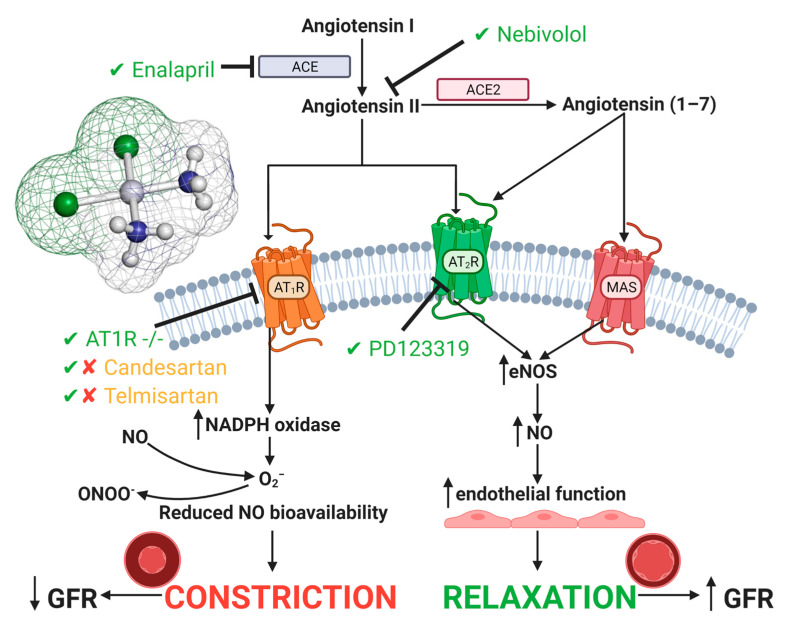
Cisplatin-induced acute kidney injury effects on the renin angiotensin system. A large focus has been on inhibition or genetic deletion of AT1 and AT2 receptors, both of which show amelioration in CIAKI [64,120,123,124,125]. Additionally, both ACE and Angiotensin II inhibition have also shown ameliorating qualities. Targeting the renin angiotensin system (RAS) system to prevent CIAKI is a promising pathway in potential treatments. Abbreviations: NO, nitric oxide; ONOO^−^, peroxynitrite; O_2_^−^, superoxide; eNOS, endothelial nitric oxide synthase; ACE, angiotensin converting enzyme; GFR, glomerular filtration rate.

**Figure 4 cancers-13-01572-f004:**
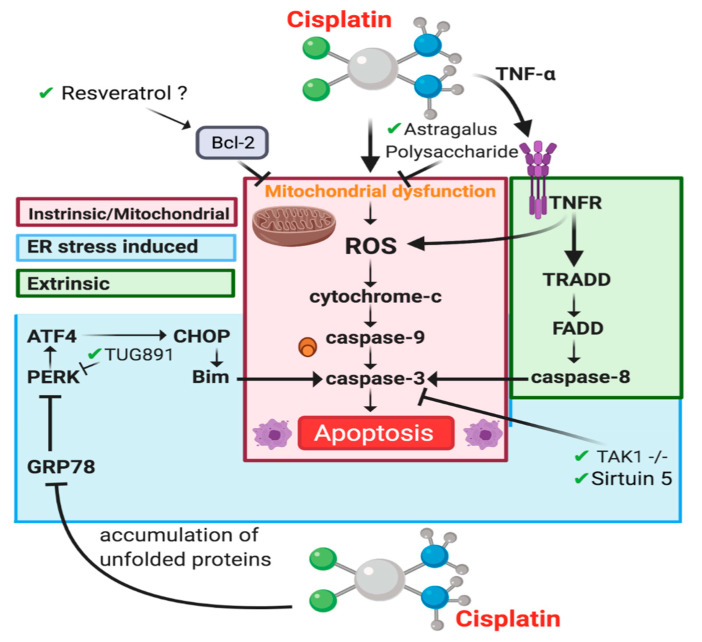
Apoptotic pathways involved in CIAKI. Cisplatin stimulation of ROS leads to cytochrome release and effector caspase activation in the Intrinsic/mitochondrial pathway (**red**) [54,94,154], ER stress is involved in two mechanisms, through stimulation of PERK to promote ER stress induced caspase-3 activation (**blue**) [155]. Extrinsic apoptosis is caused by TNFR1 signaling caspase-8 (**green**).

**Figure 5 cancers-13-01572-f005:**
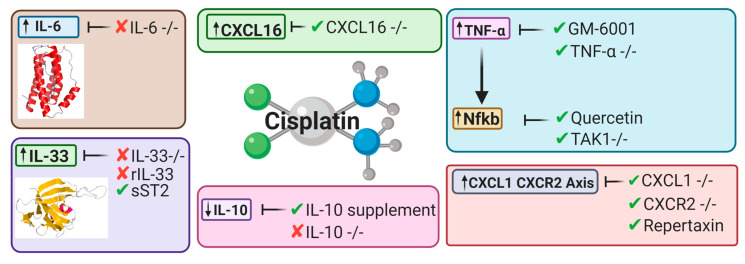
Key cytokines and chemokines upregulated or downregulated following cisplatin treatment. This diagram represents the therapeutic avenues published targeting inflammatory pathways and the targeted cytokines and chemokines involved [32,34,45,46,49,53,175,176,177].

**Figure 6 cancers-13-01572-f006:**
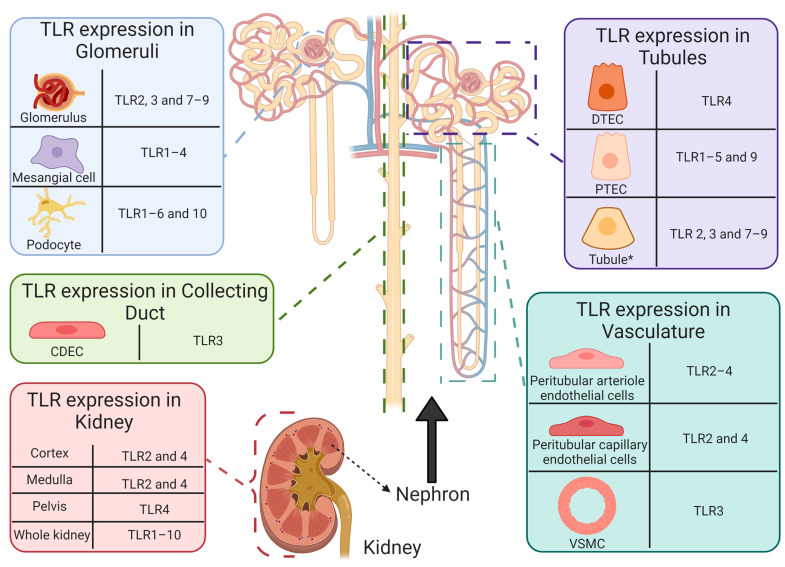
Functional expression of toll-like receptors in healthy human intrarenal cells and tissue. Basal expression of TLR1–10 has been reported in healthy human tissue [179]. However, determination of renal cell specific TLR expression remains limited. To date, TLR expression has been reported in intrarenal cells and structures, including: CDEC (TLR3 [199]), DTECs (TLR4 [200]), glomeruli (TLR2 [201]; TLR3 [199,202]; TLR7–9 [202,203]), mesangial cells (TLR1–4 [199,201]), peritubular arteriole (TLR2 [201] and TLR3 [199]) and capillary (TLR2 [201,204] and TLR4 [204]) endothelial cells, podocytes (TLR1–6 and 10 [205,206]), PTECs (TLR1–5 [204,207,208,209,210] and 9 [211]), tubules * (TLR2–4 [201,202,203,212] and TLR7–9 [203,213]), VSMCs (TLR3 [199]) and renal cortex (TLR2 [201] and TLR4 [212]), medulla (TLR2 [201] and TLR4 [212]), and pelvis (TLR4 [212]). * Tubules refers to literature that does not state the specific tubule on which TLR expression was reported. Abbreviations: collecting duct epithelial cell, CDEC; DTEC, distal tubule epithelial cell; PTEC, proximal tubule epithelial cell; toll-like receptor, TLR; vascular smooth muscle cell, VSMC.

**Table 1 cancers-13-01572-t001:** Pharmacological interventions assessed for renoprotective effects against cisplatin-induced AKI in vitro and in vivo, papers published in 2020.

Drug	Mechanism of Action	Findings	In Vitro	In Vivo	Reference
Aucubin	Anti-inflammatory	↓ Markers of oxidative stress (HO-1 and 4-HNE) ↓ Apoptosis (caspase-3, caspase-9 and PARP)	–	BALB/c mice	[38]
Curcumin	Anti-inflammatory, Antioxidative, oxygen-free radical scavenging, antifibrotic, and anticancer activities	↓ Tubular Injury ↓ BUN ↓ sCr (rats)	–	C57BL/6J mice/rats	[39,40]
Dexmedetomidine	Antiapoptotic via α2AR/PI3K/AKT pathway	↑ Body weight and renal index ↓ Tubular epithelial cell apoptosis ↓ Expression of GRP78, CHOP and Caspase-12	–	Sprague Dawley Rats	[41]
Etoricoxib	Anti-inflammatory	↓ Inflammation (iNOS) ↓ Apoptosis (BAX) No changes to creatinine, BUN, GSH, and MDA	–	Rats	[40]
Eugenol	Antioxidant and anti-inflammatory properties	↓ sCr and BUN ↓ PAS tubular injury score ↓ cytoplasmic vacuolization of proximal tubular cells	–	BALB/c mice	[42]
Ferrostatin-1	Inhibits Ferroptotic cell death	↓ sCr and BUN ↓ apoptosis (TUNEL stain) ↓ Tubular injury score (H&E) ↓ Lipid peroxidation	–	C57BL/6J mice	[43]
Isoorientin	Anti-inflammatory, antioxidant	↓ ROS generation ↓ Apoptosis ↓ Inflammation	mTECs	Nrf2^−/−^	[44]
Monotropein	Antioxidant, anti-inflammatory and antiapoptotic	↓ Tubular injury ↓ markers of oxidative stress ↓ markers of apoptosis ↓ BUN, no reduction in sCr	–	BALB/c mice	[45]
Paricalcitol	Synthetic vitamin D deficiency	↓ MDA (HK-2 cells) ↓ Cell death (HK-2 cells) ↓ sCr and BUN (WT mouse) ↓ Tissue Injury (WT mouse)	HK-2 cells	WT mice	[43]
Quercetin	Anti-inflammatory	↓ sCr and BUN ↓ mRNA expression of IL-1β, IL-6, TNF-α ↓ reduced tubular necrosis score ↓ activity of Syk/NF-κB	–	C57BL/6J mice	[46]

Abbreviations: sCr, SCr; BUN, blood urea nitrogen; PAS, periodic acid Schiff; KIM-1, kidney injury molecule-1; ROS, reactive oxygen species; mTECs, medullary thymic epithelial cells; iNOS, inducible nitric oxide synthase; BAX, BCL2-associated X protein; GSH, glutathione; MDA, malondialdehyde; TUNEL, Terminal deoxynucleotidyl transferase dUTP nick end labelling; H&E, haematoxylin and eosin; HK-2, human kidney 2; VDR, Vitamin D receptor; mRNA, messenger ribonucleic acid; ^−/−^, knockout; IL-1β, Interleukin-1 beta; IL-6, Interleukin-6; TNF-α, tumor necrosis factor-alpha; Syk, spleen tyrosine kinase; NF-κB, nuclear factor kappa B; CHOP, CCAAT-enhancer-binding protein homologous protein; GRP78, glucose-regulated protein 78; HO-1, heme oxygenase-1; 4-HNE, 4-hydroxynonenal; and PARP, poly (ADP-ribose) polymerase, ↓; Decreased, ↑; Increased, −/−; genetic deletion, −/−.

**Table 2 cancers-13-01572-t002:** Genetic deletion studies investigating in vivo mechanisms associated with inflammatory processes involved in cisplatin-induced AKI pathogenesis. Key genetic deletion studies that have been targeted in both recent and hindsight applications targeting cisplatin-induced AKI with majority focusing on inflammatory pathways.

Genetic Deletion	Mechanism of Action	Results	Knockout Model	Reference
CXCL16	Antiapoptosis and anti-inflammatory	↓ Apoptosis of tubular cells ↓ Caspase-3 activation ↓ inhibition of macrophage and T cell infiltration	CXCL16 ^−/−^ mice C57BL/6J background WT	[47]
CYP2e1	Antioxidant	↓ ROS ↓ BUN ↓ sCr ↑ creatinine clearance	CYP2e1 ^−/−^ mice 129/sv background WT	[48]
IL-6	Antioxidant	↑ 4-HNE ↓ SOD1 ↓ SOD2 (no significance) ↑ ERK phosphorylation ↑ COX-2	IL-6 ^−/−^ mice C57BL/6J background WT	[49]
IL-33	Pro-inflammatory	↑ BUN ↑ sCr ↑ NGAL No attenuation in ATN and tubular apoptosis scores ↓ tumor weight, volume, and growth ↑ Cisplatin efficacy	IL-33 ^−/−^ mice C57BL/6J background WT	[50]
NLRP3	Unknown	No change to BUN, sCr, ATN score and tubular apoptosis score.	NLRP3 ^−/−^ mice C57BL/6J background WT	[51]
PARP-1	Anti-inflammatory, antioxidant and antinitrative	↓ BUN ↓ sCr ↓ PAS tubular injury score	PARP-1 ^−/−^ mice C57BL/6J background WT	[52]
T cell	Pro-inflammatory	↑ cisplatin administration survival rate ↓ sCr ↓ tubular injury score ↓ TNF-α	nu/nu mice	[53]
TAK1	Antiapoptotic, Anti-inflammatory	↓ Apoptosis of tubular cells ↓ Caspase-3 activation ↓ reduced mRNA expression of IL-6, TNF-α, MCP-1 and MIP-2 ↓ JNK phosphorylation	PT-TAK1 ^−/−^ mice	[54]
TLR-2	Inflammatory response	↑ BUN ↑ sCr ↑ tissue injury score	TLR2 ^−/−^	[55]
TLR4	Anti-inflammatory response	↓ BUN ↓ sCr ↓ tissue injury index ↑ IL-4 and IL-10	TLR4 ^−/−^	[55]
TLR-9	Pro-inflammatory	No significant change to either serum urea or tubular injury score.	TLR-9 ^−/−^	[56]
TNF-α	Potentially anti-inflammatory	↓ BUN ↓ tubular necrosis score	TNF-α ^−/−^	[33]

Abbreviations: BUN, blood urea nitrogen; sCr, sCr; PAS, periodic acid Schiff; PARP-1, Poly (ADP-ribose) polymerase-1; TLR, Toll-like receptors; TNF-α, tumor necrosis factor-alpha; 4-HNE, 4-hydroxy-2-nonenal; SOD, Superoxide dismutase; ERK, extracellular-signal-regulated kinase; COX-2, cyclooxygenase-2; NGAL, neutrophil gelatinase-associated lipocalin; CXCL16, CXC chemokine ligand 16; TAK1, transforming growth factor b-activated kinase 1; MCP-1, monocyte chemoattractant protein-1; MIP-2, macrophage inflammatory protein 2; JNK, c-Jun N-terminal kinases; CYP2el, cytochrome P4502E1; ROS, reactive oxygen species; LRP3, LDL receptor-related protein 3; and ATN, acute tubular necrosis, ↓; Decreased, ↑; Increased, −/−; genetic deletion.

**Table 3 cancers-13-01572-t003:** Summary of location and primary pathogens recognized by toll-like receptors.

Toll-Like Receptor	Location	Primary Pathogen (s)
1	Extracellular	Gram-positive bacterium Fungus Mycobacterium
2	Extracellular	Fungus Gram-positive bacterium Mycobacterium
3	Intracellular	Double-stranded virus
4	Extracellular	Gram-negative bacterium
5	Extracellular	Flagellum
6	Extracellular	Gram-positive bacterium Fungus
7	Intracellular	Single-stranded virus
8	Intracellular	Virus
9	Intracellular	Bacterium
10	Extracellular	Gram-positive bacterium

Receptor position influences pathogen specificity, dividing toll-like receptors (TLRs) into two subgroups [195]: (a) TLR1, 2, 4, 5, 6, and 10 are anchored to the plasma membrane and are predominantly responsible for detecting components expressed on the outer surface of flagella, fungi, and Gram-negative and -positive bacteria [181,182,196], and (b) TLR3, 7–9 reside on intracellular components (including endosomes, lysosomes, and endolysosomes) and are primarily responsible for recognizing nucleic acids derived from pathogens (bacteria and viruses) [181,182]. Internalization (trafficking of cell surface receptor into internal endosome) of TLR2 [197] and TLR4 [198] has also been reported.

**Table 4 cancers-13-01572-t004:** Cisplatin-induced acute kidney injury human clinical trials.

Name of Trial	Status	Year	Results	Clinical Trial Identifier/ Reference
The Effect of Intravenous Mannitol Plus Saline on the Prevention of Cisplatin-induced Nephrotoxicity: A Randomized, Double-blind, Placebo Controlled Trial (MACIN)	Recruiting	2020	TBA	NCT04251689
The effect of melatonin on cisplatin-induced nephrotoxicity: A pilot, randomized, double-blinded, placebo-controlled clinical trial	Completed	2020	Reduced KIM-1/creatinine and NGAL/creatinine ratios indicative of reduced AKI. patients treated with melatonin showed reduced AKI episodes compared to the placebo group. This however was non-significant suggested to be due to the small sample size.	[305]
Mesenchymal Stem Cells in Cisplatin-Induced Acute Renal Failure in Patients with Solid Organ Cancers	Withdrawn	2011, updated 2018	Study withdraw as patients failed to develop acute renal failure (Key criterion for the study)	NCT01275612
Preloading Magnesium Attenuate Cisplatin-induced Nephrotoxicity	Completed	2015, updated 2019	—Beneficial in prevention of cisplatin-induced acute kidney disease No statistical significance in the prevention of CIAKI.	NCT02481518 [304]
Effects of DPP4 Inhibitor on Cisplatin-Induced Acute Kidney Injury	Unknown			NCT02250872 [306]
Preventing Nephrotoxicity and Ototoxicity from Osteosarcoma Therapy	Completed	2013, updated 2020	—serum creatinine/biomarkers of AKI (KIM-1/NGAL) were not improved by pantoprazole —The study concluded that pantoprazole was unable to ameliorate cisplatin-induced AKI.-	NCT01848457 [75]
Randomized phase II feasibility study of mannitol or furosemide hydration in moderate dose of cisplatin-based chemotherapy with short hydration for advanced non-small cell lung cancer	Completed	2014, updated 2021	No significant difference in renal toxicity compared to treatment without mannitol	UMIN000015293 [301]
Evaluation of the Effect of Acetazolamide, Mannitol and N-acetylcysteine on Cisplatin-Induced Nephrotoxicity	Completed	2016, updated 2017	TBA	NCT02760901 [307]
Effect of Silymarin Administration on Cisplatin Nephrotoxicity: Report from A Pilot, Randomized, Double-Blinded, Placebo-Controlled Clinical Trial	Completed	2013, updated 2015	NGAL/creatinine ratio was the same between silymarin and placebo groups Overall conclusion of the study is that silymarin was not effective against cisplatin-induced AKI.	[308]
Amifostine pretreatment for protection against cyclophosphamide-induced and cisplatin-induced toxicities: results of a randomized control trial in patients with advanced ovarian cancer	Completed	1996	Reduced number of patients requiring delaying or discontinue cisplatin treatment due to nephrotoxicity in Amifostine co treatment group compared to cisplatin alone. Amifostine reduced incidence of hypomagnesemia (key characteristic of cisplatin nephrotoxicity).	[309]

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
