# Peer review of "Mechanisms of Cisplatin-Induced Acute Kidney Injury: Pathological Mechanisms, Pharmacological Interventions, and Genetic Mitigations"

_cancers, 2021, doi:10.3390/cancers13071572_

Round 1

Reviewer 1 Report

The manuscript presents a well-illustarted comprehensive overview of the current knowledge of the mechanisms of cisplatin-induced nephotoxicity and approaches to prevent acute kidney injury. The review is very interesting and should be published after several issues have been adressed.

General comment

The authors should discuss the hurdles preventing the translation of the preclinical findings into clinic. They should also specify where they see the greatest potential for ameliorating cisplatin nephrotoxicity.

Minor comments

  1. Line 131: It is unclear what "cisplatin treated CTR1 knockdown human embryonic kidney (HEK293) cells". Were these cells characterized by CTR1 loss? How is this relevant for OCT2?
  2. Line 132-141. What is the link between imatinib and OCT2? Is imatinib an OCT2 substrate? 
  3. Line 219. "The cisplatin-glutathione conjugate responsible for the bioactivation of cisplatin" is not correct. On the contrary, interaction of cisplatin with GSH deactivates the platinum drug.
  4. Figure 4 is blur.

Reviewer 2 Report

The manuscript described in detail the mechanism of cisplatin-induced acute kidney injury. This is a very important subject, since cisplatin is still one of the most effective chemotherapeutics with known side effects, especially for kidneys.

I believe that the review is well organized and written, very comprehensive contemporary. The immunological part is described in detail which might be a subject of a separate review. Illustrations are adequate and clear.  I think this work will help clinicians and cancer biologists better understand this phenomenon.

Reviewer 3 Report

Manuscript “Mechanisms of cisplatin-induced acute kidney injury: Pathological mechanisms, pharmacological intervention and genetic mitigations” by Kristen McSweeney et al. has gathered large amount of literature about cisplatin induced acute kidney injury, including mechanisms involved, intervention strategies through pharmacological inhibition and genetic modified preclinical-models. It apparently a lot of effort was toward putting this together, however, manuscript needs further clarification and reorganization before considering for acceptance for publication. I have below concerns/comments in which hope authors can clarify and improve in the revision:

  1. The authors indicated that the current failure of the intervention strategies “predominately linked to the incomplete understanding of CIAKI pathophysiology and molecular mechanisms involved”, while simultaneously presented large amount of prior mechanism studies, is the intention of the review just to serve as a comprehensive summary in the field as stated “each individual mechanism linked to the disease, the pharmacological options that have been tested to target each of them and the results obtained by each study”? Which is not exactly true, eg, study involved with cellular accumulation, cell death are incomplete (will specify below).  The take home message is not quite clear, suggest authors adjust wording accordingly, and make it clear in the introduction and conclusion regarding the purpose of the review and the reasons current intervention strategies unsuccessful.
  2. Throughout the manuscript, clinical translation of the preclinical studies were imbedded in some of the section, such as section 2 in relation to OCT2 transporter, but not clearly mentioned in every sections, it would be helpful if author can summarize the emerging promising clinical trials currently ongoing in the field in a separate table. Or if the purpose of the article is only focus on preclinical findings (don’t think so, and it will lose one of most important function as a review), make it consistent throughout the manuscript.
  3. The structure of the manuscript: what’s rationale for separation of section 4 (cell death) and section 3 (molecular mechanisms)? Isn’t cell death part of molecular mechanisms?
  4. To keep section 2 (targeting cisplatin cellular uptake, efflux and accumulation) consistent, individual transporter would be better organized under category of uptake or efflux transporter, suggest to include contribution of OAT1/3 as observed in published article (Identification of OAT1/OAT3 as Contributors to Cisplatin Toxicity, PMCID: PMC5593168).
  5. Line 143/144, the statement is irrelevant and not proper for the OCT2, please justify.
  6. Section 2.6. Line 231/232 is reductant, can be deleted or moved into 2.6. the mechanism involved with MATE1 is not clearly explained, not clear why “Body weight was reduced in both MATE1 deficient and wild-type mice” helps elucidating the role of MATE1…please rewrite the paragraph.

Round 2

Reviewer 3 Report

I have reviewed the revised manuscript and think it has been significantly improved and now warrants publication in Cancers. I don't have further questions.